# On Convolutions, Intrinsic Dimension, and Diffusion Models

**Kin Kwan Leung**                                                                                      *kk@layer6.ai*
*Layer 6 AI*

**Rasa Hosseinzadeh**                                                                                   *rasa@layer6.ai*
*Layer 6 AI*

**Gabriel Loaiza-Ganem**                                                                                *gabriel@layer6.ai*
*Layer 6 AI*

**Reviewed on OpenReview:** *https://openreview.net/forum?id=xSzBf1te4s*

## Abstract

The manifold hypothesis asserts that data of interest in high-dimensional ambient spaces, such as image data, lies on unknown low-dimensional submanifolds. Diffusion models (DMs) – which operate by convolving data with progressively larger amounts of Gaussian noise and then learning to revert this process – have risen to prominence as the most performant generative models, and are known to be able to learn distributions with low-dimensional support. For a given datum in one of these submanifolds, we should thus intuitively expect DMs to have implicitly learned its corresponding local intrinsic dimension (LID), i.e. the dimension of the submanifold it belongs to. Kamkari et al. (2024b) recently showed that this is indeed the case by linking this LID to the rate of change of the log marginal densities of the DM with respect to the amount of added noise, resulting in an LID estimator known as FLIPD. LID estimators such as FLIPD have a plethora of uses, among others they quantify the complexity of a given datum, and can be used to detect outliers, adversarial examples and AI-generated text. FLIPD achieves state-of-the-art performance at LID estimation, yet its theoretical underpinnings are incomplete since Kamkari et al. (2024b) only proved its correctness under the highly unrealistic assumption of affine submanifolds. In this work we bridge this gap by formally proving the correctness of FLIPD under realistic assumptions. Additionally, we show that an analogous result holds when Gaussian convolutions are replaced with uniform ones, and discuss the relevance of this result.

## 1 Introduction

The manifold hypothesis (Bengio et al., 2013) states that high-dimensional data in $\mathbb{R}^D$ commonly lies on a low-dimensional submanifold, or a disjoint union thereof (Brown et al., 2023), embedded on $\mathbb{R}^D$; this hypothesis has been empirically found to hold for image data (Pope et al., 2021) and other modalities (Cresswell et al., 2022). Given a dataset obeying this hypothesis along with a query datum $x$, one might want to leverage the data to estimate the *local intrinsic dimension* (LID) of the datum – i.e. the dimension of the submanifold it belongs to – which we denote as $\mathtt{LID}(x)$. At a first glance the problem of LID estimation might seem like a mere statistical or mathematical curiosity, yet LID estimators have found numerous important applications within machine learning: they effectively quantify image complexity (Kamkari et al., 2024b), are helpful in the detection of outliers (Houle et al., 2018; Anderberg et al., 2024; Kamkari et al., 2024a), AI-generated text (Tulchinskii et al., 2023), memorization (Ross et al., 2025) and adversarial examples (Ma et al., 2018), and LID estimates of neural network representations are predictive of generalization (Ansuini et al., 2019; Birdal et al., 2021; Brown et al., 2022; Magai & Ayzenberg, 2022). However, traditional LID estimators rely on pairwise distances or similar quantities (Fukunaga & Olsen, 1971; Pettis et al., 1979; Grassberger & Procaccia, 1983; Levina & Bickel, 2004; MacKay & Ghahramani, 2005; Johnsson et al., 2014;

Facco et al., 2017; Bac et al., 2021), and they thus typically suffer both from poor scaling on dataset size and from the curse of dimensionality.

Generative models aim to learn the distribution $p$ that gave rise to observed data, and when the manifold hypothesis holds, $p$ has low-dimensional support. Research studying generative models through the lens of the manifold hypothesis has recently proliferated (Dai & Wipf, 2019; Zhang et al., 2020; Brehmer & Cranmer, 2020; Kim et al., 2020; Arbel et al., 2021; Caterini et al., 2021; Horvat & Pfister, 2021; Ross & Cresswell, 2021; Cunningham et al., 2022; Loaiza-Ganem et al., 2022a;b; Ross et al., 2023; Sorrenson et al., 2024; Loaiza-Ganem et al., 2024; Ventura et al., 2025), and an important finding from this line of research is that diffusion models (DMs; Sohl-Dickstein et al., 2015; Ho et al., 2020; Song et al., 2021) – which are state-of-the-art generative models – can successfully learn distributions with low-dimensional support (Pidstrigach, 2022; De Bortoli, 2022). When a generative model, such as a DM, correctly learns $p$ it must therefore learn the corresponding intrinsic dimensions of its support as well; indeed, various works have shown that LID estimators can be extracted from generative models and that these new estimators avoid the aforementioned pitfalls of traditional ones (Tempczyk et al., 2022; Zheng et al., 2022; Horvat & Pfister, 2024; Stanczuk et al., 2024; Kamkari et al., 2024b).

FLIPD is a state-of-the-art and highly scalable LID estimator based on DMs that was recently proposed by Kamkari et al. (2024b). FLIPD is grounded in a result showing that

$$\texttt{LID}(x) = D + \lim_{\delta \to -\infty} \frac{\partial}{\partial \delta} \log \varrho_{\mathcal{N}}(x, \delta), \tag{1}$$

where $\varrho_{\mathcal{N}}(\,\cdot\,, \delta)$ denotes the convolution between $p$ and Gaussian noise with log standard deviation $\delta$. DMs operate by adding Gaussian noise to data and estimating the corresponding score function; the right-hand side of Equation 1 can be written in terms of this score function, which then enables FLIPD to be computed by plugging-in the score function learned by the DM. However, Kamkari et al. (2024b) only proved Equation 1 in the case where $p$ is supported on an affine submanifold of $\mathbb{R}^D$ – this is a highly restrictive and unrealistic assumption for typical data of interest such as natural images.

In summary, LID estimation is a relevant problem in machine learning and the current best-performing estimator of LID lacks strong formal justification. In this work we remedy this situation by proving that Equation 1 actually holds when $p$ is supported on general disjoint unions of submanifolds of $\mathbb{R}^D$, thus fully justifying FLIPD. Additionally, we also prove that an analogous result holds when the convolution is carried out against a uniform distribution on a ball of log radius $\delta$ (rather than a Gaussian with log standard deviation $\delta$). Although this second result is less practically relevant since DMs do not use uniform noise, it is still of interest since it directly links $\texttt{LID}(x)$ to the probability assigned by $p$ to a ball around $x$, which is a quantity that naturally appears implicitly and explicitly throughout machine learning (see e.g. Silverman, 1986; Naeem et al., 2020; Bhattacharjee et al., 2023; Kamkari et al., 2024a).

## 2 Setup and Background

### 2.1 Setup and Notation

Throughout our work, we will follow the definition of a $d$-dimensional manifold as being locally homeomorphic to $\mathbb{R}^d$ (Lee, 2012), meaning that the dimension of a manifold does not vary within the manifold. $\mathcal{M}$ will denote, depending on context, either an embedded $d$-dimensional submanifold of $\mathbb{R}^D$, or a countable disjoint union of embedded submanifolds of potentially varying dimensions. We will assume that data of interest is supported on $\mathcal{M}$ and generated according to a probability distribution admitting a density $p : \mathcal{M} \to \mathbb{R}$.[1] Writing $\mathcal{M}$ as $\mathcal{M} = \cup_j \mathcal{M}_j$ where $\mathcal{M}_j$ is $d_j$-dimensional and $\mathcal{M}_i \cap \mathcal{M}_j = \emptyset$ for $i \neq j$, the local intrinsic dimension of $x \in \mathcal{M}$ is defined as the dimension of the manifold that $x$ belongs to, i.e. $\texttt{LID}(x) = d_i$ if $x \in \mathcal{M}_i$.

---

[1]Note that $p$ is not a density with respect to the Lebesgue measure on $\mathbb{R}^D$, but rather with respect to the Riemannian measure (or volume form if it exists) on $\mathcal{M}$; see the work of Loaiza-Ganem et al. (2024) for a discussion about this setup.

We will make heavy use of isotropic Gaussian densities with log standard deviation $\delta \in \mathbb{R}$, i.e. with covariance matrix $e^{2\delta}I_D$; for a mean $\mu \in \mathbb{R}^D$, we will denote the corresponding density evaluated at $x \in \mathbb{R}^D$ as

$$\mathcal{N}_D(x; \mu, \delta) = C_D^{\mathcal{N}} e^{-D\delta} \exp\left(-\frac{1}{2}\|x - \mu\|^2 e^{-2\delta}\right), \quad \text{where} \quad C_D^{\mathcal{N}} = (2\pi)^{-D/2}, \tag{2}$$

and where $\|\cdot\|$ will always refer to the Euclidean norm. We will sometimes find it convenient to write $\mathcal{N}_d(x; \mu, \delta)$ where $d < D$ and $x, \mu \in \mathbb{R}^D$, with the understanding that

$$\mathcal{N}_d(x; \mu, \delta) := C_d^{\mathcal{N}} e^{-d\delta} \exp\left(-\frac{1}{2}\|x - \mu\|^2 e^{-2\delta}\right) \tag{3}$$

is not a density as it does not integrate to 1. We will also heavily use the convolution of $p$ against a zero-centred Gaussian with log standard deviation $\delta$, which we denote as[2]

$$\varrho_{\mathcal{N}}(x, \delta) := \int_{\mathcal{M}} p(x') \mathcal{N}_D(x - x'; 0, \delta) \mathrm{d}x'. \tag{4}$$

We will treat uniform densities analogously to Gaussian ones, and will denote the density of a uniform random variable on a ball of log radius $\delta$ centered at $\mu \in \mathbb{R}^D$, evaluated at $x$, as

$$\mathcal{U}_D(x; \mu, \delta) = C_D^{\mathcal{U}} e^{-D\delta} \mathbb{1}(x \in B_D(\mu, e^{\delta})), \quad \text{where} \quad C_D^{\mathcal{U}} = \pi^{-D/2} \Gamma\left(\frac{D}{2} + 1\right), \tag{5}$$

and where $\mathbb{1}(\cdot)$ denotes an indicator function, $B_D(\mu, r)$ denotes a $D$-dimensional ball of radius $r$ centred at $\mu$, i.e. $B_D(\mu, r) := \{x \in \mathbb{R}^D : \|x - \mu\| < r\}$, and $\Gamma$ is the gamma function. In analogy to the Gaussian case, we will also use

$$\mathcal{U}_d(x; \mu, \delta) := C_d^{\mathcal{U}} e^{-d\delta} \mathbb{1}(x \in B_D(\mu, e^{\delta})) \tag{6}$$

for $x, \mu \in \mathbb{R}^D$, and will also write

$$\varrho_{\mathcal{U}}(x, \delta) := \int_{\mathcal{M}} p(x') \mathcal{U}_D(x - x'; 0, \delta) \mathrm{d}x'. \tag{7}$$

As mentioned earlier, we will also prove Equation 1 when $\varrho_{\mathcal{N}}(x, \delta)$ is replaced by $\varrho_{\mathcal{U}}(x, \delta)$. Since

$$\varrho_{\mathcal{U}}(x, \delta) = \int_{\mathcal{M} \cap B_D(x, e^{\delta})} p(x') C_D^{\mathcal{U}} e^{-D\delta} \mathrm{d}x' = C_D^{\mathcal{U}} e^{-D\delta} \mathbb{P}_{X \sim p}(X \in B_D(x, e^{\delta})), \tag{8}$$

our result thus explicitly links $\mathrm{LID}(x)$ to the probability assigned by $p$ to a ball of log radius $\delta$ around $x$ through

$$\mathrm{LID}(x) = \lim_{\delta \to -\infty} \frac{\partial}{\partial \delta} \log \mathbb{P}_{X \sim p}(X \in B_D(x, e^{\delta})). \tag{9}$$

## 2.2 Diffusion Models and FLIPD

As already mentioned, the main goal of our paper is to prove Equation 1 under general and realistic assumptions. In the rest of this section we cover the background that makes this equation relevant, namely DMs and FLIPD. Our proofs do not rely on the content presented in this section, which can safely be skipped by readers who are already familiar with these topics, or by those who are only interested in our theoretical results. We will follow the stochastic differential equation (SDE) formulation of DMs of Song et al. (2021). DMs are generative models whose goal is to learn $p$; they do this by first instantiating a (forward) noising process through an Itô SDE,

$$\mathrm{d}X_t := \alpha(X_t, t)\mathrm{d}t + \beta(t)\mathrm{d}W_t, \quad X_0 \sim p, \tag{10}$$

where $\alpha : \mathbb{R}^D \times [0, 1] \to \mathbb{R}^D$ and $\beta : [0, 1] \to \mathbb{R}$ are fixed functions which are specified as hyperparameters, and $W_t$ is a $D$-dimensional Brownian motion. We will denote the density of $X_t$ implied by Equation 10

---

[2]Note that, since $p$ is not a density with respect to the Lebesgue measure, the integral in Equation 4 is not a Lebesgue integral, and $\mathrm{d}x'$ should be understood as the Riemannian measure or volume form on $\mathcal{M}$. Nonetheless, $\varrho_{\mathcal{N}}(\cdot, \delta)$ is a Lebesgue density in $\mathbb{R}^D$ due to the added Gaussian noise.

as $p(\,\cdot\,,t)$, and note that $p(\,\cdot\,,0) = p$.[3] Here $t \in [0,1]$ corresponds to an artificial time variable specifying how much noise has been added to the data through the SDE, with $t = 0$ corresponding to no noise, and with $t = 1$ corresponding to $p(\,\cdot\,,1)$ being "almost pure noise". A surprising result by Anderson (1982) and Haussmann & Pardoux (1986) shows that the reverse process, $Y_t \coloneqq X_{1-t}$, also obeys a (backward) SDE,

$$\mathrm{d}Y_t = \left[\beta^2(1-t)s(Y_t, 1-t) - \alpha(Y_t, 1-t)\right]\mathrm{d}t + \beta(1-t)\mathrm{d}\hat{W}_t, \quad Y_0 \sim p(\,\cdot\,,1), \tag{11}$$

where $s$ is the (Stein) score function, i.e. $s(x,t) \coloneqq \nabla \log p(x,t)$, and $\hat{W}_t$ is another $D$-dimensional Brownian motion. Equation 11 is at the core of DMs: by approximating the score function $s$ with a properly trained neural network $\hat{s}$ (and also approximating $p(\,\cdot\,,1)$ with an appropriately chosen Gaussian distribution), numerically solving the resulting SDE until time $t = 1$ will produce samples from a DM – these samples will be approximately distributed according to $p$; the better the approximations made throughout, the closer the distribution of these samples will be to $p$ (De Bortoli, 2022).

The function $\alpha$ is often chosen as $\alpha(x,t) = \gamma(t)x$ for some function $\gamma : [0,1] \to \mathbb{R}$. Under this choice, the transition kernel corresponding to Equation 10 is known (Särkkä & Solin, 2019), and it is given by

$$p_{t|0}(x_t \mid x_0) = \mathcal{N}_D(x_t; \psi(t)x_0, \log\sigma(t)), \tag{12}$$

where the functions $\psi, \sigma : [0,1] \to \mathbb{R}$ are uniquely determined by $\gamma$ and $\beta$, and can be numerically evaluated for all common choices of $\gamma$ and $\beta$. Furthermore, under these standard choices the ratio $\lambda(t) \coloneqq \sigma(t)/\psi(t)$ is injective and therefore admits a left inverse $\lambda^{-1}$, which can also be numerically evaluated. Recall that the defining property of the transition kernel is that it satisfies

$$p(x,t) = \int_{\mathcal{M}} p(x')p_{t|0}(x \mid x')\mathrm{d}x'. \tag{13}$$

The density in Equation 13 resulting from the SDE in Equation 10 is extremely similar to the convolution in Equation 4: both of them operate by adding Gaussian noise to data from $p$ (up to the rescaling of the mean by $\psi(t)$ in Equation 12). The main difference between these two ways of adding noise is simply how the amount of added noise is measured: in the DM formulation, no noise corresponds to $t = 0$, whereas this setting corresponds to the limit as $\delta \to -\infty$ for convolutions. Intuitively, it should then be the case that if one has access to a DM, then the right hand side of Equation 1 can be approximated, thus obtaining an estimate of LID. Kamkari et al. (2024b) used this intuition to propose FLIPD. First, they showed that $\varrho_{\mathcal{N}}(\,\cdot\,,\delta)$ and $p(\,\cdot\,,t)$ are related through

$$\log\varrho_{\mathcal{N}}(x,\delta) = D\log\psi\big(t(\delta)\big) + \log p\Big(\psi\big(t(\delta)\big)x, t(\delta)\Big), \tag{14}$$

where $t(\delta) \coloneqq \lambda^{-1}(e^\delta)$. Then, using the Fokker-Planck equation – which provides an explicit formula for $\frac{\partial}{\partial t}p(x,t)$ – along with the chain rule and Equation 14, Kamkari et al. (2024b) showed that

$$\frac{\partial}{\partial\delta}\log\varrho_{\mathcal{N}}(x,\delta) = \sigma^2\big(t(\delta)\big)\left(\mathrm{Tr}\left(\nabla s\Big(\psi\big(t(\delta)\big)x, t(\delta)\Big)\right) + \left\|s\Big(\psi\big(t(\delta)\big)x, t(\delta)\Big)\right\|^2\right) =: \nu\big(s, x, t(\delta)\big). \tag{15}$$

Importantly, evaluating $\nu$ requires access to $p(x,t)$ only through $s$. Lastly, by using the trained score function $\hat{s}$ to approximate the unknown $s$ and by setting a negative enough $\delta_0$, FLIPD is obtained by combining Equation 15 with Equation 1 as

$$\mathtt{FLIPD}\,(x;\delta_0) \coloneqq D + \nu\big(\hat{s}, x, t(\delta_0)\big). \tag{16}$$

The correctness of FLIPD as an estimator of LID therefore hinges on Equation 1 holding in general, and as previously mentioned, Kamkari et al. (2024b) only proved Equation 1 in the case where $\mathcal{M}$ is an affine submanifold of $\mathbb{R}^D$. We refer the reader to Kamkari et al. (2024b) for a more thorough derivation of FLIPD, along with illustrative examples, and empirical results.

---

[3]Note that $p(\,\cdot\,,0)$ and $p(\,\cdot\,,t)$ for $t > 0$ are not densities in the same sense; only the latter are Lebesgue densities.

## 3    Related Work

As mentioned in the introduction, statistical estimators of LID typically rely on the computation of all pairwise distances or angles on a given dataset sampled from $p$ (Fukunaga & Olsen, 1971; Pettis et al., 1979; Grassberger & Procaccia, 1983; Levina & Bickel, 2004; MacKay & Ghahramani, 2005; Johnsson et al., 2014; Facco et al., 2017; Bac et al., 2021), resulting in these estimators exhibiting poor scaling on dataset size and ambient dimension $D$. More related to our work is research aiming to leverage deep generative models for LID estimation. Stanczuk et al. (2024) proposed an estimator which also leverages DMs but is not based on Equation 1 and requires many more function evaluations of $\hat{s}$ than FLIPD. Horvat & Pfister (2024) proposed another method using DMs for LID estimation, but it requires altering the training procedure of the DM. As a result, these methods are not compatible with off-the-shelf state-of-the-art DMs such as Stable Diffusion (Rombach et al., 2022) – to the best of our knowledge FLIPD is the only estimator which scales to this setting, making it particularly relevant to properly justify it.

Other works have used generative models beyond DMs, for example Zheng et al. (2022) showed that the number of active latent dimensions in variational autoencoders (Kingma & Welling, 2014; Rezende et al., 2014) estimates LID. Lastly, Tempczyk et al. (2022) proved another convolution-related result which they leveraged for LID estimation with normalizing flows (Dinh et al., 2017; Durkan et al., 2019). More specifically, they showed that for $x \in \mathcal{M}$, as $\delta \to -\infty$,

$$\log \varrho_{\mathcal{N}}(x, \delta) = \delta(\texttt{LID}(x) - D) + \mathcal{O}(1). \tag{17}$$

By adding different levels of noise $\delta$ to data and training a normalizing flow for each noise level, Tempczyk et al. (2022) estimate $\log \varrho_{\mathcal{N}}(x, \delta)$ for various values of $\delta$; they then fit a linear regression predicting these estimates from $\delta$ – the resulting slope is an estimate $\texttt{LID}(x) - D$ thanks to Equation 17. When proposing FLIPD, Kamkari et al. (2024b) noted that Equation 17 provides informal intuition for why Equation 1 holds: if the $\mathcal{O}(1)$ term was constant with respect to $\delta$, then Equation 17 would imply Equation 1. In this sense, proving Equation 1 boils down to showing that the $\mathcal{O}(1)$ term indeed behaves like a constant as $\delta \to -\infty$, or more formally, that the limit of its derivative with respect to $\delta$ converges to 0.

Several works have studied the interplay between DMs and the manifold hypothesis in contexts beyond LID estimation: Pidstrigach (2022) first showed that if the error between $\hat{s}$ and $s$ is bounded, then DMs recover the correct support, thus showing that DMs can learn manifolds. De Bortoli (2022) refined this result with error bounds in Wasserstein distance, and various follow-up works have provided finite-sample estimation rates (Chen et al., 2023; Oko et al., 2023; Tang & Yang, 2024; Wang et al., 2024). Lastly, Chen et al. (2024) observed that, under the manifold hypothesis, the Jacobian of the score function has low rank, and they further leveraged this observation for image editing.

## 4    Result Statements and Discussion

In this section we state and discuss our results. Partial proofs that avoid differential geometry (Lee, 2012; 2018) are provided in Section 5; the remaining steps, which rely on these tools, are given in the appendices. In our first result, we establish that Equation 1 holds when $\mathcal{M}$ is an arbitrary submanifold of $\mathbb{R}^D$ and $p$ obeys continuity and finite second moment conditions.

**Theorem 1.** *Let $\mathcal{M}$ be a smooth $d$-dimensional embedded submanifold of $\mathbb{R}^D$ and let $p$ be a probability density function on $\mathcal{M}$. Let $x \in \mathcal{M}$ be such that $p$ is continuous at $x$, $p(x) > 0$, and $C := \int_{\mathcal{M}} p(x') \texttt{dist}_{\mathcal{M}}^2(x, x') \mathrm{d}x' < \infty$, where $\texttt{dist}_{\mathcal{M}}$ denotes the geodesic distance on $\mathcal{M}$ obtained from the induced Riemannian metric on $\mathcal{M}$. Then,*

$$\lim_{\delta \to -\infty} \frac{\partial}{\partial \delta} \log \varrho_{\mathcal{N}}(x, \delta) = d - D. \tag{18}$$

Our continuity and second moment assumptions in Theorem 1 are mild and are completely analogous to corresponding assumptions made by Kamkari et al. (2024b). Nevertheless, we emphasize once again that Kamkari et al. (2024b) also assumed that $\mathcal{M}$ is an affine submanifold of $\mathbb{R}^D$, which is a much stronger and unrealistic assumption on the structure of $\mathcal{M}$. Watchful readers might have noticed that the finite second

moment assumption – i.e. $\int_{\mathcal{M}} p(x') \mathtt{dist}_{\mathcal{M}}^2(x, x') \mathrm{d}x' < \infty$ – implies that $\mathcal{M}$ must be connected (or more precisely, $p$ must assign all its probability mass to a single connected component of $\mathcal{M}$), which is likely not realistic in many settings of practical interest. However this obstacle is minor since, as we show below, Theorem 1 can be straightforwardly extended to the case where $\mathcal{M}$ is a disjoint union of submanifolds of $\mathbb{R}^D$ of potentially varying dimensions.

**Corollary 1.** *Let $\mathcal{M} = \cup_j \mathcal{M}_j$, where $\mathcal{M}_j$ is a smooth $d_j$-dimensional embedded submanifold of $\mathbb{R}^D$ with $\xi := \min_{i \neq j} \inf_{x_i \in \mathcal{M}_i, x_j \in \mathcal{M}_j} \|x_i - x_j\| > 0$. Let $p$ be a probability density function on $\mathcal{M}$, which we write as $p(x) = \pi_j p_j(x)$ for $x \in \mathcal{M}_j$, where for every $j$, $p_j$ is a probability density on $\mathcal{M}_j$ and $\pi_j > 0$, and $\sum_j \pi_j = 1$.[4] Let $x \in \mathcal{M}_i$ for some $i$ be such that $p_i$ is continuous at $x$, $p_i(x) > 0$, and $\int_{\mathcal{M}_i} p_i(x') \mathtt{dist}_{\mathcal{M}_i}^2(x, x') \mathrm{d}x' < \infty$, where $\mathtt{dist}_{\mathcal{M}_i}$ denotes the geodesic distance on $\mathcal{M}_i$ obtained from the induced Riemannian metric on $\mathcal{M}_i$. Then,*

$$\lim_{\delta \to -\infty} \frac{\partial}{\partial \delta} \log \varrho_{\mathcal{N}}(x, \delta) = d_i - D. \tag{19}$$

Corollary 1 formally justifies FLIPD in the sense that it shows that, under mild regularity conditions,

$$\lim_{\delta \to -\infty} D + \nu(s, x, t(\delta)) = \mathtt{LID}(x), \tag{20}$$

that is, when the score function is perfectly learned, i.e. $\hat{s} = s$, we have that $\mathtt{FLIPD}(x; \delta) \to \mathtt{LID}(x)$ as $\delta \to -\infty$.

For the uniform case, analogously to the Gaussian one, we first prove the case where $\mathcal{M}$ is a $d$-dimensional submanifold, and then extend the result to disjoint unions of submanifolds. Note that our results with uniform distributions do not require the second moment conditions required in the Gaussian case.

**Theorem 2.** *Let $\mathcal{M}$ be a smooth $d$-dimensional embedded submanifold of $\mathbb{R}^D$ and let $p$ be a probability density function on $\mathcal{M}$. Let $x \in \mathcal{M}$ be such that $p$ is continuous at $x$ and $p(x) > 0$. Then,*

$$\lim_{\delta \to -\infty} \frac{\partial}{\partial \delta} \log \varrho_{\mathcal{U}}(x, \delta) = d - D. \tag{21}$$

**Corollary 2.** *Let $\mathcal{M} = \cup_j \mathcal{M}_j$, where $\mathcal{M}_j$ is a smooth $d_j$-dimensional embedded submanifold of $\mathbb{R}^D$ with $\xi := \min_{i \neq j} \inf_{x_i \in \mathcal{M}_i, x_j \in \mathcal{M}_j} \|x_i - x_j\| > 0$. Let $p$ be a probability density function on $\mathcal{M}$, which we write as $p(x) = \pi_j p_j(x)$ for $x \in \mathcal{M}_j$, where for every $j$, $p_j$ is a probability density on $\mathcal{M}_j$ and $\pi_j > 0$, and $\sum_j \pi_j = 1$. Let $x \in \mathcal{M}_i$ be such that $p_i$ is continuous at $x$ and $p_i(x) > 0$. Then,*

$$\lim_{\delta \to -\infty} \frac{\partial}{\partial \delta} \log \varrho_{\mathcal{U}}(x, \delta) = d_i - D. \tag{22}$$

As previously discussed, Corollary 2 immediately links $\mathtt{LID}(x)$ and $\mathbb{P}_{X \sim p}(X \in B_D(x, e^\delta))$ through Equation 9. Albeit Corollary 1 is more relevant due to its connection to DMs and FLIPD, Corollary 2 still succeeds at connecting two quantities of interest in machine learning despite lacking these connections. Some older estimators of LID use a given a dataset sampled from $p$ by regressing the log proportion of datapoints in $B_D(x, e^\delta)$ against $\delta$, and then estimating $\mathtt{LID}(x)$ as the corresponding slope (Pettis et al., 1979; Grassberger & Procaccia, 1983). These works provided an intuitively correct but not rigorous mathematical justification behind this estimator, which Equation 9 rigorously justifies.

## 5 Proofs

### 5.1 Gaussian Case: Theorem 1 and Corollary 1

*Proof of Theorem 1.* The core idea behind the proof of Theorem 1 is to exploit the following relationship between $\mathcal{N}_D(x; \mu, \delta)$ and $\mathcal{N}_d(x; \mu, \delta)$:

$$\mathcal{N}_D(x - x'; 0, \delta) = (2\pi)^{\frac{d-D}{2}} e^{\delta(d-D)} \mathcal{N}_d(x - x'; 0, \delta). \tag{23}$$

---

[4]Note that, because $\xi > 0$, without loss of generality any density $p$ on $\mathcal{M}$ can be written as $p(x) = \pi_j p_j(x)$ when $x \in \mathcal{M}_j$.

Thus we have

$$\varrho_{\mathcal{N}}(x, \delta) = \int_{\mathcal{M}} p(x')\mathcal{N}_D(x - x'; 0, \delta)\mathrm{d}x' = (2\pi)^{\frac{d-D}{2}} e^{\delta(d-D)} \int_{\mathcal{M}} p(x')\mathcal{N}_d(x - x'; 0, \delta)\mathrm{d}x'. \tag{24}$$

To simplify notation, we let

$$p_\delta^{\mathcal{N}}(x) := \int_{\mathcal{M}} p(x')\mathcal{N}_d(x - x'; 0, \delta)\mathrm{d}x', \tag{25}$$

so that

$$\frac{\partial}{\partial \delta} \log \varrho_{\mathcal{N}}(x, \delta) = d - D + \frac{\partial}{\partial \delta} \log p_\delta^{\mathcal{N}}(x). \tag{26}$$

Thus it suffices to show that

$$\lim_{\delta \to -\infty} \frac{\partial}{\partial \delta} \log p_\delta^{\mathcal{N}}(x) = 0. \tag{27}$$

First, note that we have

$$\frac{\partial}{\partial \delta}\mathcal{N}_d(x - x'; 0, \delta) = (-d + \|x - x'\|^2 e^{-2\delta})\mathcal{N}_d(x - x'; 0, \delta). \tag{28}$$

Then,

$$\lim_{\delta \to -\infty} \frac{\partial}{\partial \delta} \log p_\delta^{\mathcal{N}}(x) = \lim_{\delta \to -\infty} \frac{\frac{\partial}{\partial \delta} p_\delta^{\mathcal{N}}(x)}{p_\delta^{\mathcal{N}}(x)} \tag{29}$$

$$= \lim_{\delta \to -\infty} \frac{\frac{\partial}{\partial \delta} \int_{\mathcal{M}} p(x')\mathcal{N}_d(x - x'; 0, \delta)\mathrm{d}x'}{\int_{\mathcal{M}} p(x')\mathcal{N}_d(x - x'; 0, \delta)\mathrm{d}x'} \tag{30}$$

$$= \lim_{\delta \to -\infty} \frac{\int_{\mathcal{M}} p(x')\frac{\partial}{\partial \delta}\mathcal{N}_d(x - x'; 0, \delta)\mathrm{d}x'}{\int_{\mathcal{M}} p(x')\mathcal{N}_d(x - x'; 0, \delta)\mathrm{d}x'} \tag{31}$$

$$= \lim_{\delta \to -\infty} \frac{\int_{\mathcal{M}} p(x')(-d + \|x - x'\|^2 e^{-2\delta})\mathcal{N}_d(x - x'; 0, \delta)\mathrm{d}x'}{\int_{\mathcal{M}} p(x')\mathcal{N}_d(x - x'; 0, \delta)\mathrm{d}x'} \tag{32}$$

$$= -d + \lim_{\delta \to -\infty} \frac{e^{-2\delta} \int_{\mathcal{M}} p(x')\|x - x'\|^2 \mathcal{N}_d(x - x'; 0, \delta)\mathrm{d}x'}{\int_{\mathcal{M}} p(x')\mathcal{N}_d(x - x'; 0, \delta)\mathrm{d}x'}. \tag{33}$$

Note that going from Equation 30 to Equation 31 involves moving the derivative inside the integral. We formally justify this step in Appendix A. We then leverage the following two propositions, whose proofs we also include in Appendix A.

**Proposition 1.** *Under the assumptions of Theorem 1,*

$$\lim_{\delta \to -\infty} \int_{\mathcal{M}} p(x')\mathcal{N}_d(x - x'; 0, \delta)\mathrm{d}x' = p(x). \tag{34}$$

Note that Proposition 1 is very similar to the limit of a Gaussian being a delta function, i.e. for a function $f : \mathbb{R}^d \to \mathbb{R}$ and $y \in \mathbb{R}^d$, we have that $\int_{\mathbb{R}^d} f(y')\mathcal{N}_d(y - y'; 0, \delta)\mathrm{d}y' \to f(y)$ as $\delta \to -\infty$. However, the differences here are that: ($i$) we are computing the convolution on a manifold $\mathcal{M}$, and ($ii$) the distance in the "Gaussian" is the distance of the ambient space, $\mathbb{R}^D$.

**Proposition 2.** *Under the assumptions of Theorem 1,*

$$\lim_{\delta \to -\infty} e^{-2\delta} \int_{\mathcal{M}} p(x')\|x - x'\|^2 \mathcal{N}_d(x - x'; 0, \delta)\mathrm{d}x' = d \cdot p(x). \tag{35}$$

The expected squared norm of a centred Gaussian is well-known to be the trace of its covariance, i.e. $\int_{\mathbb{R}^d} \|y - y'\|^2 \mathcal{N}_d(y - y'; 0, \delta)\mathrm{d}y' = de^{2\delta}$. Intuitively, Proposition 2 can be understood as stating that this property still holds as $\delta \to -\infty$ under ($i$) and ($ii$), with $p(x')$ again leaving the integral as $p(x)$ because the Gaussian behaves like a delta function in this regime.

Applying these two propositions to Equation 33 yields Equation 27, which in turn finishes the proof of Theorem 1. $\qquad\square$

*Proof of Corollary 1.* Under the conditions of Corollary 1, $\varrho_{\mathcal{N}}(x, \delta)$ is given by

$$\varrho_{\mathcal{N}}(x, \delta) = \sum_j \pi_j \int_{\mathcal{M}_j} p_j(x')\mathcal{N}_D(x - x'; 0, \delta)\mathrm{d}x'. \tag{36}$$

Following an analogous derivation to the one leading to Equation 33, we get

$$\lim_{\delta \to -\infty} \frac{\partial}{\partial \delta} \log \varrho_{\mathcal{N}}(x, \delta) = -D + \lim_{\delta \to -\infty} \frac{\sum_j \pi_j \int_{\mathcal{M}_j} p_j(x')\|x - x'\|^2 e^{-2\delta}\mathcal{N}_D(x - x'; 0, \delta)\mathrm{d}x'}{\sum_j \pi_j \int_{\mathcal{M}_j} p_j(x')\mathcal{N}_D(x - x'; 0, \delta)\mathrm{d}x'}. \tag{37}$$

Using basic calculus to maximize $\mathcal{N}_D(x - x'; 0, \delta)$ and $\|x - x'\|^2 e^{-2\delta}\mathcal{N}_D(x - x'; 0, \delta)$ with respect to $\|x - x'\|$ subject to $\|x - x'\| \geq \xi$ yields that, when this constraint holds,

$$\mathcal{N}_D(x - x'; 0, \delta) \leq C_D^{\mathcal{N}} e^{-D\delta} \exp(-\tfrac{1}{2}\xi^2 e^{-2\delta}), \tag{38}$$

and that if additionally $\delta \leq \frac{1}{2}\log\frac{\xi}{2}$, then

$$\|x - x'\|^2 e^{-2\delta}\mathcal{N}_D(x - x'; 0, \delta) \leq \xi^2 e^{-2\delta} C_D^{\mathcal{N}} e^{-D\delta} \exp(-\tfrac{1}{2}\xi^2 e^{-2\delta}). \tag{39}$$

The bound in Equation 38 decreases monotonically to 0 as $\delta \to -\infty$, and the bound in Equation 39 does so as well provided that $\delta < \log\xi - \frac{1}{2}\log(D + 2)$ (which makes the derivative of the log of the bound with respect to $\delta$ positive). It follows that $\mathcal{N}_D(x - x'; 0, \delta)$ and $\|x - x'\|^2 e^{-2\delta}\mathcal{N}_D(x - x'; 0, \delta)$ are both uniformly bounded over $x'$ and $\delta$ when $x' \in \mathcal{M}_j$ for $j \neq i$ and $\delta < \min(\frac{1}{2}\log\frac{\xi}{2}, \log\xi - \frac{1}{2}\log(D + 2))$. The dominated convergence theorem then guarantees that, for $j \neq i$:

$$\lim_{\delta \to -\infty} \int_{\mathcal{M}_j} p_j(x')\mathcal{N}_D(x - x'; 0, \delta)\mathrm{d}x' = 0 = \lim_{\delta \to -\infty} \int_{\mathcal{M}_j} p_j(x')\|x - x'\|^2 e^{-2\delta}\mathcal{N}_D(x - x'; 0, \delta)\mathrm{d}x'. \tag{40}$$

Lastly, since $x \in \mathcal{M}_i$, Proposition 1 ensures that $\lim_{\delta \to -\infty} \int_{\mathcal{M}_i} p_i(x')\mathcal{N}_D(x - x'; 0, \delta)\mathrm{d}x' > 0$, so that Equation 37 reduces to

$$\lim_{\delta \to -\infty} \frac{\partial}{\partial \delta} \log \varrho_{\mathcal{N}}(x, \delta) = -D + \lim_{\delta \to -\infty} \frac{\int_{\mathcal{M}_i} p_i(x')\|x - x'\|^2 e^{-2\delta}\mathcal{N}_D(x - x'; 0, \delta)\mathrm{d}x'}{\int_{\mathcal{M}_i} p_i(x')\mathcal{N}_D(x - x'; 0, \delta)\mathrm{d}x'} \tag{41}$$

$$= \lim_{\delta \to -\infty} \frac{\partial}{\partial \delta} \log \int_{\mathcal{M}_i} p_i(x')\mathcal{N}_D(x - x'; 0, \delta)\mathrm{d}x' = d_i - D, \tag{42}$$

where the last equality follows from Theorem 1. $\qquad\square$

## 5.2 Uniform Case: Theorem 2 and Corollary 2

*Proof of Theorem 2 (informal).* In analogy with Theorem 1, the proof of Theorem 2 hinges on the following relationship between $\mathcal{U}_D(x; \mu, \delta)$ and $\mathcal{U}_d(x; \mu, \delta)$:

$$\mathcal{U}_D(x; \mu, \delta) = C_D^{\mathcal{U}}(C_d^{\mathcal{U}})^{-1} e^{\delta(d - D)}\mathcal{U}_d(x; \mu, \delta). \tag{43}$$

Thus we have

$$\varrho_{\mathcal{U}}(x, \delta) = \int_{\mathcal{M}} p(x')\mathcal{U}_D(x - x'; 0, \delta)\mathrm{d}x' = C_D^{\mathcal{U}}(C_d^{\mathcal{U}})^{-1} e^{\delta(d - D)} \int_{\mathcal{M}} p(x')\mathcal{U}_d(x - x'; 0, \delta)\mathrm{d}x'. \tag{44}$$

To simplify notation once again, we let

$$p_\delta^{\mathcal{U}}(x) := \int_{\mathcal{M}} p(x')\mathcal{U}_d(x - x'; 0, \delta)\mathrm{d}x', \tag{45}$$

so that

$$\frac{\partial}{\partial \delta} \log \varrho_{\mathcal{U}}(x, \delta) = d - D + \frac{\partial}{\partial \delta} \log p_\delta^{\mathcal{U}}(x). \tag{46}$$

Thus it suffices to show that

$$\lim_{\delta \to -\infty} \frac{\partial}{\partial \delta} \log p_\delta^{\mathcal{U}}(x) = 0. \tag{47}$$

Despite seeming extremely similar to the Gaussian case from Equation 27, the proof of Equation 47 is not completely analogous. We now provide some intuition as to why this equation holds. Note that $p_\delta^{\mathcal{U}}(x) = C_d^{\mathcal{U}} e^{-d\delta} \mathbb{P}_{X \sim p}(X \in B_D(x, e^\delta))$. Intuitively, since $p$ is continuous at $x$ it can be locally approximated by a constant, meaning that when $\delta \to -\infty$, we should expect $\mathbb{P}_{X \sim p}(X \in B_D(x, e^\delta))$ to behave like a constant times the volume (in $\mathcal{M}$) of $\mathcal{M} \cap B_D(x, e^\delta)$. Since $\mathcal{M}$ is $d$-dimensional, we should also expect $\mathcal{M} \cap B_D(x, e^\delta)$ to behave as a $d$-dimensional ball as $\delta \to -\infty$, i.e. to have a volume proportional to $e^{d\delta}$. Putting these intuitions together, we get that $p_\delta^{\mathcal{U}}(x)$ behaves like a constant as $\delta \to -\infty$, thus yielding Equation 47.

We formalize the above intuition in Appendix B by providing a rigorous proof of Equation 47 – this result completes the proof of Theorem 2. $\qquad\square$

*Proof of Corollary 2.* Under the assumptions of Corollary 2, $\varrho_{\mathcal{U}}(x, \delta)$ is given by

$$\varrho_{\mathcal{U}}(x, \delta) = \sum_j \pi_j \int_{\mathcal{M}_j} p_j(x') \mathcal{U}_D(x - x'; 0, \delta) \mathrm{d}x'. \tag{48}$$

Whenever $\delta < \log \xi$, if $x' \in \mathcal{M}_j$ for $j \neq i$, we have that $\mathcal{U}_D(x - x'; 0, \delta) = 0$ since $x \in \mathcal{M}_i$, in which case

$$\varrho_{\mathcal{U}}(x, \delta) = \pi_i \int_{\mathcal{M}_i} p_i(x') \mathcal{U}_D(x - x'; 0, \delta) \mathrm{d}x'. \tag{49}$$

By Theorem 2, it follows that

$$\lim_{\delta \to -\infty} \frac{\partial}{\partial \delta} \log \varrho_{\mathcal{U}}(x, \delta) = \lim_{\delta \to -\infty} \frac{\partial}{\partial \delta} \log \int_{\mathcal{M}_i} p_i(x') \mathcal{U}_D(x - x'; 0, \delta) \mathrm{d}x' = d_i - D. \tag{50}$$

$\qquad\square$

# 6 Conclusions and Future Work

In this paper we established a formal link between LID and the rate of change of the densities of different convolutions. The Gaussian case is of particular interest as it provides formal justification for FLIPD, a state-of-the-art LID estimator. The uniform case provides an interesting connection between LID and the probability assigned to Euclidean balls, and rigorously justifies older LID estimators.

One avenue for future work would be extending our result beyond Gaussian and uniform distributions. For example, despite losing direct relevance in machine learning, finding sufficient conditions on the convolving distributions for our result to hold is an interesting mathematical problem. In particular, we hypothesize that, under adequate regularity conditions, Theorem 1 and Theorem 2 can be generalized to cases where the noise distribution that $p$ is convolved against obeys a decomposition like those of Equation 23 or Equation 43.

Another interesting avenue for future work is extending LID estimation to flow matching methods (Lipman et al., 2023; Albergo & Vanden-Eijnden, 2023; Liu et al., 2023; Tong et al., 2024). These methods aim to train a vector field $\hat{v}$ such that solving the ordinary differential equation

$$\mathrm{d}Y_t = \hat{v}(Y_t, 1 - t)\mathrm{d}t, \quad Y_0 \sim p(\,\cdot\,, 1) \tag{51}$$

results in $Y_1$ being distributed according to the data distribution $p(\,\cdot\,, 0)$. Flow matching methods differ in how they obtain such a $\hat{v}$. For example, when using the so called linear interpolation method (Lipman et al., 2023), $\hat{v}$ is equivalent to an affine transformation of a score function $\hat{s}$ (see Appendix D.3 in Kingma & Gao (2023)), which means that FLIPD can be trivially applied in this setting. However, not all flow matching methods produce a $\hat{v}$ from which one can easily recover a corresponding score function $\hat{s}$ to plug into FLIPD; for example we should expect this to be the case when using several rounds of rectified flows (Liu et al., 2023) or optimal transport regularization (Tong et al., 2024). We believe that producing a tractable estimate of

$\text{LID}(x)$ while given access only to a successfully trained $\hat{v}$ from any flow matching method is an interesting research problem.

Lastly, although our results indeed justify FLIPD, they do so in the setting where the score function is perfectly learned, i.e. $\hat{s} = s$. Despite DMs being highly performant, they never perfectly recover the true score function, and discrepancies between $\hat{s}$ and $s$ can result in

$$\lim_{\delta \to -\infty} \text{FLIPD}\,(x; \delta) = \text{LID}(x) \tag{52}$$

being violated. Indeed, Kamkari et al. (2024b) reported that in some instances, FLIPD can produce negative estimates of LID. Since Kamkari et al. (2024b) only proved Equation 1 when $\mathcal{M}$ is an affine submanifold of $\mathbb{R}^D$, the possibility remained that this negativity was due to the result not holding more generally; a consequence of Corollary 1 is that negative estimates can only be caused by $\hat{s}$ having imperfectly learned $s$. We believe that quantifying how much $\text{FLIPD}\,(x; \delta_0)$ differs from $\text{LID}(x)$ as a function of the error incurred by $\hat{s}$ to approximate $s$ is another relevant problem. In particular, note that if we had bounds on both the error between $s$ and $\hat{s}$, and on the error between $\nabla s$ and $\nabla \hat{s}$, as $\delta \to -\infty$, we could trivially obtain a bound on $|\nu(s, x, t(\delta)) - \nu(\hat{s}, x, t(\delta))|$ (see Equation 15); in turn, this would provide a bound on the LID estimation error. While we are aware of learning theory work bounding the error between $s$ and $\hat{s}$ (Chen et al., 2023; Oko et al., 2023; Tang & Yang, 2024; Wang et al., 2024), we are unaware of any existing bound on the error between $\nabla s$ and $\nabla \hat{s}$; finding such a bound is another interesting direction for future work.

### Acknowledgments

We thank Hamidreza Kamkari for useful conversations during the early stages of this research.

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

## A    Proofs for the Gaussian Case

The following claim provides justification for taking the derivative inside the integral from Equation 30 to Equation 31.

**Claim 1.** *Under the assumptions of Theorem 1,*

$$\frac{\partial}{\partial \delta} \int_{\mathcal{M}} p(x') \mathcal{N}_d(x - x'; 0, \delta) \mathrm{d}x' = \int_{\mathcal{M}} p(x') \frac{\partial}{\partial \delta} \mathcal{N}_d(x - x'; 0, \delta) \mathrm{d}x'. \tag{53}$$

*Proof.* Let $\delta \in \mathbb{R}$, and let $\delta_a$ be such that $\delta_a < \delta$. We begin by bounding $\frac{\partial}{\partial \delta} \mathcal{N}_d(x - x'; 0, \delta)$:

$$|\frac{\partial}{\partial \delta} \mathcal{N}_d(x - x'; 0, \delta)| = |-d + \|x - x'\|^2 e^{-2\delta} |C_d^{\mathcal{N}} e^{-d\delta} \exp\left(-\frac{1}{2} \|x - x'\|^2 e^{-2\delta}\right) \tag{54}$$

$$\leq (d + \|x - x'\|^2 e^{-2\delta_a}) C_d^{\mathcal{N}} e^{-d\delta_a}. \tag{55}$$

This bound is integrable due to the finite second moment assumption:

$$\int_{\mathcal{M}} p(x') (d + \|x - x'\|^2 e^{-2\delta_a}) C_d^{\mathcal{N}} e^{-d\delta_a} \mathrm{d}x' \leq \int_{\mathcal{M}} p(x') (d + \mathtt{dist}_{\mathcal{M}}(x, x')^2 e^{-2\delta_a}) C_d^{\mathcal{N}} e^{-d\delta_a} \mathrm{d}x' < \infty. \tag{56}$$

The result then follows from the Leibniz integral rule. □

In the rest of this section we prove Proposition 1 and Proposition 2. Note that we share notation through the rest of the section, i.e. some of the notation we introduce in the proof of Proposition 1 is also used after this proof.

*Proof of Proposition 1.* Recall that we want to prove that

$$\lim_{\delta \to -\infty} \int_{\mathcal{M}} p(x') \mathcal{N}_d(x - x'; 0, \delta) \mathrm{d}x' = p(x). \tag{57}$$

To compute this limit, without loss of generality, we perform a translation so that $x = 0$. Note that this translation preserves distance. Then perform an orthogonal transformation on $\mathbb{R}^D$ such that the tangent plane of $\mathcal{M}$ at $x = 0$ is the $d$-subspace spanned by the first $d$ coordinate vectors. Denote these coordinates by $(y_1, \ldots, y_d)$. As $\mathcal{M}$ is smooth, $\mathcal{M}$ can be parametrized by $\{y, u(y)\}$ near $x = 0$, where $y = (y_1, \ldots, y_d)$ and $u$ is a function from some open $U \subset \mathbb{R}^d$ to $\mathbb{R}^{D-d}$, such that $u(0) = 0$ and $(Du)(0) = 0$.[5] In other words, $(y_1, \ldots, y_d)$ is a local coordinate system of $\mathcal{M}$ on $U$, with the induced Riemannian metric $g$. Note that $g = I_d$ at $x = 0$ as $(Du)(0) = 0$. This means that $U$ is diffeomorphic to an open set in $\mathcal{M}$ via $\phi(x) = (x, u(x))$. Shrink $U$ if necessary such that $U$ is bounded, i.e. $R := \sup_{x' \in \phi(U)} \|x'\| < \infty$. Further shrink $U$ if necessary such that $B_D(0, R) \cap \mathcal{M} = \phi(U)$. In other words, $x' \in \mathcal{M}, \|x'\| < R \iff x' \in \phi(U)$.

We are going to compute the limit on some open $V \subset \phi(U)$ in $\mathcal{M}$ and $\mathcal{M} \setminus V$. Let $R_V := \sup_{x' \in V} \|x'\| < \infty$. Then for $x' \in \mathcal{M} \setminus V$, for any $a \in \mathbb{R}$, we have

$$e^{a\delta} \mathcal{N}_d(-x'; 0, \delta) = (2\pi)^{-d/2} e^{-\delta(d-a)} \exp\left(-\frac{1}{2} \|x'\|^2 e^{-2\delta}\right) \tag{58}$$

$$\leq (2\pi)^{-d/2} e^{-\delta(d-a)} \exp\left(-\frac{1}{2} R_V^2 e^{-2\delta}\right), \tag{59}$$

which approaches zero as $\delta \to -\infty$. This means that for any $\epsilon > 0$, there exists $K_{V,a}$ such that $\delta < K_{V,a}$ implies $e^{a\delta} \mathcal{N}_d(-x'; 0, \delta) < \epsilon$.

---

[5]To avoid using non-standard notation for derivatives, we overload notation and use $D$ to refer to both the derivative operator and the ambient dimension. The meaning of $D$ will always be clear from context.

If $x' \in V$, in the coordinate system $(y_1, \ldots, y_d)$, we have $\mathcal{N}_d(-x'; 0, \delta)$ represented by

$$\hat{N}(y, \delta) := (2\pi)^{-d/2} e^{-\delta d} \exp\left(-\frac{1}{2}(\|y\|^2 + \|u(y)\|^2)e^{-2\delta}\right). \tag{60}$$

For simplicity, let

$$N(y, \delta) := (2\pi)^{-d/2} e^{-\delta d} \exp\left(-\frac{1}{2}\|y\|^2 e^{-2\delta}\right). \tag{61}$$

Note that $N(y, \delta)$ is the standard multivariate gaussian distribution with covariance $e^{2\delta} I_d$.

It is clear that $\hat{N}(y, \delta) \leq N(y, \delta)$. Now, consider the function $v(y) = \frac{\|u(y)\|}{\|y\|}$. Note that $v(y)$ is continuous everywhere in $U \setminus \{0\}$. Since $u$ and the derivatives of $u$ vanish at $y = 0$, from the definition of derivative, we have

$$0 = \lim_{y \to 0} \frac{\|u(y) - u(0) - (Du)(0)y\|}{\|y\|} = \lim_{y \to 0} \frac{\|u(y)\|}{\|y\|} = \lim_{y \to 0} v(y).$$

Thus $v(y)$ can be extended to a continuous function in $U$.

Now let $K_V = \max_{y \in \phi^{-1}(\overline{V})} v(y)$. Then we have $\|u(y)\|^2 \leq K_V^2 \|y\|^2$. Thus

$$\hat{N}(y, \delta) = (2\pi)^{-d/2} e^{-\delta d} \exp\left(-\frac{1}{2}(\|y\|^2 + \|u(y)\|^2)e^{-2\delta}\right) \tag{62}$$

$$\geq (2\pi)^{-d/2} e^{-\delta d} \exp\left(-\frac{1}{2}\|y\|^2(1 + K_V^2)e^{-2\delta}\right) \tag{63}$$

$$= e^{d\delta_0}(2\pi)^{-d/2} e^{-(\delta+\delta_0)d} \exp\left(-\frac{1}{2}\|y\|^2 e^{-2(\delta+\delta_0)}\right) \tag{64}$$

$$= (1 + K_V^2)^{-d/2} N(y, \delta + \delta_0), \tag{65}$$

where $\delta_0 = -\frac{1}{2}\log(1 + K_V^2)$.

To summarize, we have

$$(1 + K_V^2)^{-d/2} N(y, \delta + \delta_0) \leq \hat{N}(y, \delta) \leq N(y, \delta). \tag{66}$$

Now, for any open $V \subset \phi(U)$ in $\mathcal{M}$, we have

$$p_\delta(0) = \int_{\mathcal{M}} p(x') \mathcal{N}_d(-x'; 0, \delta) \mathrm{d}x' \tag{67}$$

$$= \int_{\mathcal{M} \setminus V} p(x') \mathcal{N}_d(-x'; 0, \delta) \mathrm{d}x' + \int_V p(x') \mathcal{N}_d(-x'; 0, \delta) \mathrm{d}x'. \tag{68}$$

For any $\epsilon > 0$, whenever $\delta < K_{V,0}$ we have,

$$\int_{\mathcal{M} \setminus V} p(x') N_d(-x'; 0, \delta) \mathrm{d}x' \leq \int_{\mathcal{M} \setminus V} p(x') \epsilon \mathrm{d}x' \leq \epsilon \int_{\mathcal{M}} p(x') \mathrm{d}x' = \epsilon, \tag{69}$$

as $p$ is a probability density on $\mathcal{M}$. This shows that

$$\lim_{\delta \to -\infty} \int_{\mathcal{M} \setminus V} p(x') \mathcal{N}_d(-x'; 0, \delta) \mathrm{d}x' = 0. \tag{70}$$

Note that this is true for *any* open $V \subset \phi(U)$ in $M$.

On the other hand, in the local coordinate system $(y_1, \ldots, y_d)$, we have

$$\int_V p(x') \mathcal{N}_d(-x'; 0, \delta) \mathrm{d}x' = \int_{\phi^{-1}(V)} p(y) \sqrt{g(y)} \hat{N}(y, \delta) \mathrm{d}y. \tag{71}$$

Here $g(y) = \det(g_{ij}(y))$ is continuous on $\phi^{-1}(V)$ with $g(0) = 1$. From Equation 66, we have

$$(1 + K_V^2)^{-d/2} \int_{\phi^{-1}(V)} p(y)\sqrt{g(y)}N(y, \delta + \delta_0)\mathrm{d}y \tag{72}$$

$$\leq \int_{\phi^{-1}(V)} p(y)\sqrt{g(y)}\hat{N}(y, \delta)\mathrm{d}y \tag{73}$$

$$\leq \int_{\phi^{-1}(V)} p(y)\sqrt{g(y)}N(y, \delta)\mathrm{d}y. \tag{74}$$

Since $N(y, \delta)$ is the usual multivariate Gaussian distribution, it converges to the delta function. Thus, taking $\delta \to -\infty$, we have

$$(1+K_V^2)^{-d/2}p(0) \leq \liminf_{\delta\to-\infty} \int_{\phi^{-1}(V)} p(y)\sqrt{g(y)}\hat{N}(y, \delta)\mathrm{d}y \leq \limsup_{\delta\to-\infty} \int_{\phi^{-1}(V)} p(y)\sqrt{g(y)}\hat{N}(y, \delta)\mathrm{d}y \leq p(0), \tag{75}$$

since $\sqrt{g(0)} = 1$. But we have

$$\liminf_{\delta\to-\infty} p_\delta^{\mathcal{N}}(0) = \liminf_{\delta\to-\infty} \int_{\mathcal{M}} p(x')\mathcal{N}_d(-x'; 0, \delta)\mathrm{d}x' \tag{76}$$

$$= \lim_{\delta\to-\infty} \int_{\mathcal{M}\backslash V} p(x')\mathcal{N}_d(-x'; 0, \delta)\mathrm{d}x' + \liminf_{\delta\to-\infty} \int_{V} p(x')\mathcal{N}_d(-x'; 0, \delta)\mathrm{d}x' \tag{77}$$

$$= \liminf_{\delta\to-\infty} \int_{V} p(x')\mathcal{N}_d(-x'; 0, \delta)\mathrm{d}x' \tag{78}$$

$$= \liminf_{\delta\to-\infty} \int_{\phi^{-1}(V)} p(y)\sqrt{g(y)}\hat{N}(y, \delta)\mathrm{d}y, \tag{79}$$

and similarly for $\limsup$. This means that

$$(1 + K_V^2)^{-d/2}p(0) \leq \liminf_{\delta\to-\infty} p_\delta^{\mathcal{N}}(0) \leq \limsup_{\delta\to-\infty} p_\delta^{\mathcal{N}}(0) \leq p(0). \tag{80}$$

Note that these inequalities hold for all $V \subset \phi(U)$. First notice that the function $f(z) := (1 + z^2)^{-d/2}$ is continuous at $z = 0$, that $f(0) = 1$, and that $f(z) \leq 1$. For any $\epsilon > 0$, these exists $\kappa > 0$ such that whenever $|z| < \kappa$, we have $1 - (1 + z^2)^{-d/2} < \epsilon$. As $v(y)$ is continuous at $y = 0$ and $v(0) = 0$, there exists $\kappa' > 0$ such that whenever $\|y\| < \kappa'$, we have $v(y) < \kappa/2$. Thus let $V := \phi(B_d(0, \kappa') \cap U)$, we have $K_V \leq \kappa/2 < \kappa$ and thus $(1 + K_V^2)^{-d/2} > 1 - \epsilon$. Thus we have

$$p(0)(1 - \epsilon) \leq \liminf_{\delta\to-\infty} p_\delta^{\mathcal{N}}(0) \leq \limsup_{\delta\to-\infty} p_\delta^{\mathcal{N}}(0) \leq p(0) \tag{81}$$

for every $\epsilon > 0$. This shows that

$$\lim_{\delta\to-\infty} p_\delta^{\mathcal{N}}(0) = p(0). \tag{82}$$

$\square$

Before proving Proposition 2, we need to establish some basic tools.

**Claim 2.** *Let $\overline{B_d(0, r)}$ be the closed ball in $\mathbb{R}^d$ centred at 0 of radius $r$. Then*

$$\lim_{\delta\to-\infty} \int_{\mathbb{R}^d\backslash\overline{B_d(0,r)}} \|y\|^2 e^{-2\delta}N(y, \delta)\mathrm{d}y = 0. \tag{83}$$

*Proof.* Without loss of generality, assume $\delta < \log r - \frac{1}{2}\log(d + 2)$, then the integrand will decrease as $\delta$ decreases (the derivative of the log of the integrand with respect to $\delta$ is positive) and it converges pointwise to 0. It follows that the integrand is bounded for negative enough $\delta$, and the result follows from the dominated convergence theorem. $\square$

**Claim 3.** *Let $W$ be an open and bounded subset of $\mathbb{R}^d$ with $0 \in W$. Let $f : W \to \mathbb{R}$ continuous at $0$ and bounded on $W$. Then we have*

$$\lim_{\delta \to -\infty} \int_W f(y)\|y\|^2 e^{-2\delta} N(y,\delta)\mathrm{d}y = f(0)d. \tag{84}$$

*Proof.* For simplicity, let $H(y,\delta) = \|y\|^2 e^{-2\delta} N(y,\delta)$. First we note that $N(y,\delta)$ is the density of multivariate Gaussian distribution with covariance matrix $e^{2\delta}I_d$. This means that

$$\int_{\mathbb{R}^d} \|y\|^2 N(y,\delta)\mathrm{d}y = \mathbb{E}_{\mathcal{N}_d(y;0,\delta)}[\|y\|^2] = \mathrm{Tr}(e^{2\delta}I_d) = de^{2\delta}. \tag{85}$$

As $f$ is continuous at $0$, for any $\epsilon > 0$, there exists $r > 0$ such that $\|y\| < r \Rightarrow |f(y) - f(0)| < \epsilon$. Let $K$ be an upper bound of $f$ in $W$. Using Claim 2, we can pick $K'$ such that when $\delta < K'$ we have $\int_{\mathbb{R}^d \backslash \overline{B_d(0,r)}} H(y,\delta) < \epsilon$. Then for $\delta < K'$, we have

$$\left| \int_W f(y)H(y,\delta)\mathrm{d}y - f(0)d \right| \tag{86}$$

$$= \left| \int_W f(y)H(y,\delta)\mathrm{d}y - \int_{\mathbb{R}^d} f(0)H(y,\delta)\mathrm{d}y \right| \tag{87}$$

$$= \left| \int_{\overline{B_d(0,r)}} (f(y) - f(0))H(y,\delta)\mathrm{d}y + \int_{W \backslash \overline{B_d(0,r)}} f(y)H(y,\delta)\mathrm{d}y - \int_{\mathbb{R}^d \backslash \overline{B_d(0,r)}} f(0)H(y,\delta)\mathrm{d}y \right| \tag{88}$$

$$< \epsilon \int_{\overline{B_d(0,r)}} H(y,\delta)\mathrm{d}y + K \int_{W \backslash \overline{B_d(0,r)}} H(y,\delta)\mathrm{d}y + |f(0)| \int_{\mathbb{R}^d \backslash \overline{B_d(0,r)}} H(y,\delta)\mathrm{d}y \tag{89}$$

$$\leq \epsilon \int_{\mathbb{R}^d} H(y,\delta)\mathrm{d}y + K \int_{\mathbb{R}^d \backslash \overline{B_d(0,r)}} H(y,\delta)\mathrm{d}y + |f(0)| \int_{\mathbb{R}^d \backslash \overline{B_d(0,r)}} H(y,\delta)\mathrm{d}y \tag{90}$$

$$\leq d\epsilon + K\epsilon + |f(0)|\epsilon \tag{91}$$

$$= \epsilon(d + K + |f(0)|). \tag{92}$$

This proves the claim. $\qquad \square$

*Proof of Proposition 2.* Recall that we want to prove that

$$\lim_{\delta \to -\infty} e^{-2\delta} \int_{\mathcal{M}} p(x')\|x - x'\|^2 \mathcal{N}_d(x - x';0,\delta)\mathrm{d}x' = d \cdot p(x). \tag{93}$$

We perform the same translation and orthogonal transformation as in the proof of Proposition 1, which ensures that $x = 0$ and that the tangent plane of $\mathcal{M}$ at $x = 0$ is the $d$-subspace spanned by the first $d$ coordinate vectors. We once again consider an open $V \subset \phi(U)$ in $\mathcal{M}$ such that $0 \in V$, and break the integral on $V$ and on $\mathcal{M} \backslash V$.

For any $\epsilon > 0$, whenever $\delta < K_{V,-2}$ we have,

$$\int_{\mathcal{M} \backslash V} p(x')\|x'\|^2 e^{-2\delta} \mathcal{N}_d(-x';0,\delta)\mathrm{d}x' \leq \int_{\mathcal{M} \backslash V} p(x')\|x'\|^2 \epsilon \mathrm{d}x' \tag{94}$$

$$\leq \epsilon \int_{\mathcal{M} \backslash V} p(x')\mathrm{dist}_{\mathcal{M}}^2(x',0)\mathrm{d}x' \tag{95}$$

$$\leq \epsilon \int_{\mathcal{M}} p(x')\mathrm{dist}_{\mathcal{M}}^2(x',0)\mathrm{d}x' \tag{96}$$

$$= C\epsilon. \tag{97}$$

Here we used the assumption that the expected squared distance from $x = 0$ is finite. This shows that

$$\lim_{\delta \to -\infty} \int_{\mathcal{M} \backslash V} p(x')\|x'\|^2 e^{-2\delta} \mathcal{N}_d(-x';0,\delta)\mathrm{d}x' = 0. \tag{98}$$

Note that this is true for *any* open $V \subset \phi(U)$ containing $x = 0$ in $\mathcal{M}$. On the other hand, in the local coordinate system $(y_1, \ldots, y_d)$, we have

$$\int_V p(x')\|x'\|^2 \mathcal{N}_d(-x'; 0, \delta)\mathrm{d}x' = \int_{\phi^{-1}(V)} p(y)\sqrt{g(y)}(\|y\|^2 + \|u(y)\|^2)\hat{N}(y, \delta)\mathrm{d}y. \tag{99}$$

By Claim 3, we have that

$$e^{-2\delta} \int_{\phi^{-1}(V)} p(y)\sqrt{g(y)}(\|y\|^2 + \|u(y)\|^2)\hat{N}(y, \delta)\mathrm{d}y \tag{100}$$

$$= e^{-2\delta} \int_{\phi^{-1}(V)} p(y)\sqrt{g(y)}\|y\|^2(1 + v(y)^2)\hat{N}(y, \delta)\mathrm{d}y \tag{101}$$

$$\leq e^{-2\delta} \int_{\phi^{-1}(V)} p(y)\sqrt{g(y)}\|y\|^2(1 + v(y)^2)N(y, \delta)\mathrm{d}y \tag{102}$$

$$\rightarrow p(0)\sqrt{g(0)}(1 + v(0)^2)d \tag{103}$$

$$= p(0)d \tag{104}$$

as $\delta \to -\infty$. Here we use the fact that $f(y) := p(y)\sqrt{g(y)}(1 + v(y))$ is bounded, because $\phi^{-1}(\overline{V})$ is compact. The lower bound will be

$$e^{-2\delta} \int_{\phi^{-1}(V)} p(y)\sqrt{g(y)}(\|y\|^2 + \|u(y)\|^2)\hat{N}(y, \delta)\mathrm{d}y \tag{105}$$

$$= e^{-2\delta} \int_{\phi^{-1}(V)} p(y)\sqrt{g(y)}\|y\|^2(1 + v(y)^2)\hat{N}(y, \delta)\mathrm{d}y \tag{106}$$

$$\geq (1 + K_V^2)^{-d/2} e^{-2\delta} \int_{\phi^{-1}(V)} p(y)\sqrt{g(y)}\|y\|^2(1 + v(y)^2)N(y, \delta + \delta_0)\mathrm{d}y \tag{107}$$

$$\rightarrow (1 + K_V^2)^{-d/2} p(0)\sqrt{g(0)}(1 + v(0)^2)d \tag{108}$$

$$= (1 + K_V^2)^{-d/2} p(0)d \tag{109}$$

as $\delta \to -\infty$. As in the proof of Proposition 1, we can bound the limit inferior and limit superior and shrink $V$ as we like, thus finishing the proof. $\square$

## B   Proofs for the Uniform Case

*Proof of Theorem 2.* Recall that it is enough to show that

$$\lim_{\delta \to -\infty} \frac{\partial}{\partial \delta} \log p_\delta^{\mathcal{U}}(x) = 0. \tag{110}$$

We begin by performing the same rotation and translation as in Appendix A, and considering the same coordinate vectors, parameterization of $\mathcal{M}$, and open set $U$. Note that we have

$$p_\delta^{\mathcal{U}}(0) = \int_{\mathcal{M}} p(x')\mathcal{U}_d(-x'; 0, \delta)\mathrm{d}x' = C_d^{\mathcal{U}} e^{-d\delta} \int_{\mathcal{M} \cap B_D(0, e^\delta)} p(x')\mathrm{d}x'. \tag{111}$$

Assume that $\delta$ is negative enough for $\mathcal{M} \cap B_D(x, e^\delta) \subset \phi(U)$, which is open in $\mathcal{M}$. Thus, in the coordinate system $(y_1, \ldots, y_d)$, we have

$$p_\delta^{\mathcal{U}}(0) = C_d^{\mathcal{U}} e^{-d\delta} \int_{V_\delta} p(y)\sqrt{g(y)}\mathrm{d}y, \tag{112}$$

where $V_\delta := \phi^{-1}(\mathcal{M} \cap B_D(x, e^\delta)) = \{y : \|y\|^2 + \|u(y)\|^2 < e^{2\delta}\}$, $g(y) = \det(g_{ij}(y))$ is continuous on $V_\delta$, and $g(0) = 1$. Now, we have

$$\frac{\partial}{\partial \delta} \log p_\delta^{\mathcal{U}}(0) = \frac{1}{p_\delta^{\mathcal{U}}(0)} \frac{\partial}{\partial \delta} p_\delta^{\mathcal{U}}(0) \tag{113}$$

$$= \frac{1}{p_\delta^{\mathcal{U}}(0)} \left( -d p_\delta^{\mathcal{U}}(0) + C_d^{\mathcal{U}} e^{-d\delta} \frac{\partial}{\partial \delta} \int_{V_\delta} p(y) \sqrt{g(y)} \mathrm{d}y \right) \tag{114}$$

$$= -d + \frac{\frac{\partial}{\partial \delta} \int_{V_\delta} p(y) \sqrt{g(y)} \mathrm{d}y}{\int_{V_\delta} p(y) \sqrt{g(y)} \mathrm{d}y}. \tag{115}$$

So, we must prove that

$$\lim_{\delta \to -\infty} \frac{\frac{\partial}{\partial \delta} \int_{V_\delta} p(y) \sqrt{g(y)} \mathrm{d}y}{\int_{V_\delta} p(y) \sqrt{g(y)} \mathrm{d}y} = d. \tag{116}$$

Now, let $A_\delta := \frac{\partial}{\partial \delta} \int_{V_\delta} \mathrm{d}y$ be the derivative of $Vol(V_\delta)$ with respect to $\delta$. Since $p\sqrt{g}$ is continuous at $y = 0$, we have

$$\frac{\partial}{\partial \delta} \int_{V_\delta} p(y) \sqrt{g(y)} \mathrm{d}y = \lim_{h \to 0} \frac{1}{h} \left( \int_{V_{\delta+h}} p(y) \sqrt{g(y)} \mathrm{d}y - \int_{V_\delta} p(y) \sqrt{g(y)} \mathrm{d}y \right) \tag{117}$$

$$= \lim_{h \to 0} \frac{1}{h} \int_{V_{\delta,h}} p(y) \sqrt{g(y)} \mathrm{d}y, \tag{118}$$

where $V_{\delta,h} = \{y : e^{2\delta} < \|y\|^2 + \|u(y)\|^2 < e^{2(\delta+h)}\}$. In what follows we will consider the case where $h > 0$; the case where $h < 0$ is analogous. Note that for small enough $e^{2\delta}$ and $h$, $V_{\delta,h}$ is diffeomorphic to a $d$-dimensional annulus, as $U$ is a local chart for $\phi(U)$, and for small enough open set, the intersection of the ball in $\mathbb{R}^D$ with $\mathcal{M}$ is diffeomorphic to the ball in $\mathbb{R}^d$.

Now for any $\epsilon > 0$, there exists $\kappa > 0$ such that $|p(y)\sqrt{g(y)} - p(0)| < \epsilon$ whenever $\|y\| < \kappa$, since $g(0) = 1$. Choose $K'$ such that when $\delta < K'$ we have $V_{\delta,h} \subset B_d(0, \kappa)$. Note that a priori $K'$ depends on $h$. But since we are interested in the behaviour as $h \to 0$, we can choose $K'$ so that the condition holds for $h < h_0$ for some $h_0 > 0$. This means that for $\delta < K'$, we have

$$\left| \frac{1}{h} \int_{V_{\delta,h}} (p(y)\sqrt{g(y)} - p(0)) \mathrm{d}y \right| \leq \frac{1}{h} \int_{V_{\delta,h}} |p(y)\sqrt{g(y)} - p(0)| \mathrm{d}y < \frac{\epsilon}{h} \int_{V_{\delta,h}} \mathrm{d}y. \tag{119}$$

This means that

$$\frac{p(0) - \epsilon}{h} \int_{V_{\delta,h}} \mathrm{d}y < \frac{1}{h} \int_{V_{\delta,h}} p(y) \sqrt{g(y)} \mathrm{d}y < \frac{p(0) + \epsilon}{h} \int_{V_{\delta,h}} \mathrm{d}y. \tag{120}$$

Taking $h \to 0$, we have

$$(p(0) - \epsilon)A_\delta \leq \frac{\partial}{\partial \delta} \int_{V_\delta} p(y) \sqrt{g(y)} \mathrm{d}y \leq (p(0) + \epsilon)A_\delta. \tag{121}$$

Note that a similar result as above for the same $\kappa$ and $K'$ applies to the denominator of Equation 116, i.e. without the derivatives,

$$(p(0) - \epsilon)Vol(V_\delta) < \int_{V_\delta} p(y) \sqrt{g(y)} \mathrm{d}y < (p(0) + \epsilon)Vol(V_\delta). \tag{122}$$

Putting Equation 121 and Equation 122 together, we get that, for $\delta < K'$:

$$\frac{(p(0) - \epsilon)A_\delta}{(p(0) + \epsilon)Vol(V_\delta)} < \frac{\frac{\partial}{\partial \delta} \int_{V_\delta} p(y) \sqrt{g(y)} \mathrm{d}y}{\int_{V_\delta} p(y) \sqrt{g(y)} \mathrm{d}y} < \frac{(p(0) + \epsilon)A_\delta}{(p(0) - \epsilon)Vol(V_\delta)}. \tag{123}$$

To calculate $A_\delta$, we use the Leibniz integral rule in its most general form (Flanders, 1973),

$$A_\delta = \frac{\partial}{\partial \delta} \int_{V_\delta} \mathrm{d}y = \int_{V_\delta} \iota_{\dot{\Phi}} \mathrm{d}\mathrm{d}y + \int_{\partial V_\delta} \iota_{\dot{\Phi}} \mathrm{d}y + \int_{V_\delta} \dot{\mathrm{d}}y = \int_{\partial V_\delta} \iota_{\dot{\Phi}} \mathrm{d}y, \tag{124}$$

as $\mathrm{d}y$ is closed and independent of $\delta$. Here $\iota_{\dot{\Phi}}$ denotes the interior product with $\dot{\Phi}$, which is the vector field of the velocity, i.e. if $\Phi_\delta$ is a family of diffeormorphisms from a fixed domain $V$ to $V_\delta$, then $\dot{\Phi}$ is the derivative of $\Phi_\delta$ with respect to $\delta$.

Thus, in what follows we construct a family of diffeomorphisms from a fixed domain to $V_\delta$ so that we can then apply the Leibniz integral rule. We first establish some ground work for $V_\delta$. Note that $V_\delta = \{y : G(y) < \delta\}$, where $G(y) := \frac{1}{2} \log(\|y\|^2 + \|u(y)\|^2)$. Then we have

$$\frac{\partial G}{\partial y_i} = \frac{y_i + \sum_j u_j u_{j,i}}{\|y\|^2 + \|u\|^2}, \tag{125}$$

where we use the notation that any indices after a comma indicate a derivative, i.e. $u_{j,i} = \frac{\partial u_j}{\partial y_i}$. This means

$$\nabla G = \frac{y + (Du)u}{\|y\|^2 + \|u\|^2}. \tag{126}$$

**Proposition 3.** *There exists an open set $W$ containing $0$ such that $\nabla G$ is well-defined and non-vanishing on $W \setminus \{0\}$.*

To show this proposition, first we establish the following facts:

**Claim 4.** *Let $f : \mathbb{R}^d \to \mathbb{R}$ be smooth such that $f(0) = 0$. Then $f(y) = \sum_i y_i g_i(y)$ for some smooth functions $g_i$.*

*Proof.* Note that

$$f(y) = f(y) - f(0) = \int_0^1 \frac{\partial}{\partial t} f(ty) dt = \int_0^1 \sum_i y_i \frac{\partial f}{\partial y_i} f(ty) dt. \tag{127}$$

Thus $g_i = \int_0^1 \frac{\partial f}{\partial y_i} f(ty) dt$ and $g_i$ is smooth. $\qquad \square$

**Corollary 3.** *With $u(y)$ as defined above, we have $u(y) = \sum_{i,j} A_{ij}(y) y_i y_j$ for some smooth functions $A_{ij} : \mathbb{R}^d \to \mathbb{R}^{D-d}$.*

*Proof.* Since $u(0) = 0$, we have $u_k : \mathbb{R}^d \to \mathbb{R}$ such that $u_k(0) = 0$. Thus, by Claim 4, $u_k(y) = \sum_j g_{kj}(y) y_j$ for some smooth $g_{kj}$. Since all the derivatives of $u$ vanish at $0$, for any $k, l$, we have $0 = u_{k,l}(0) = g_{kl}(0) + \sum_j g_{kj,l}(0) \cdot 0 = g_{kl}(0)$. Thus, once again by Claim 4, $g_{kl}(y) = \sum_j h_{klj}(y) y_j$ for smooth $h_{klj}$. Thus we have $u_k(y) = \sum_i g_{ki}(y) y_i = \sum_{i,j} h_{kij}(y) y_i y_j$. Thus $(A_{ij})_k = h_{kij}$, which completes the proof. $\qquad \square$

**Claim 5.** *The functions $\frac{u^\top u}{y^\top y}$, $\frac{y^\top (Du)u}{y^\top y}$, and $\frac{u^\top (Du)^\top (Du)u}{y^\top y}$ can be extended to $C^1$ functions on $V_\delta$, such that each function and their first derivatives vanish at $y = 0$.*

*Proof.* Note that since $u$ and $\|y\|^2$ are smooth, the only point at which we have to prove the above functions are $C^1$ is at $y = 0$. Using Einstein summation notation, we first notice that since $u_j = A_{jkl} y_k y_l$ from Corollary 3, we have

$$u_{j,i} = A_{jkl,i} y_k y_l + A_{jki} y_k + A_{jil} y_l, \tag{128}$$

which is a polynomial in $y_i$ of degree at least 1.

For $u^\top u$, we have

$$u^\top u = u_i u_i = A_{ijk} A_{ilm} y_j y_k y_l y_m, \tag{129}$$

where every term in this polynomial in $y_i$ is of degree at least 4.

For $y^\top (Du)u$, we have

$$y^\top (Du)u = y_i u_{j,i} u_j = A_{jkl} y_i y_k y_l (A_{jkl,i} y_k y_l + A_{jki} y_k + A_{jil} y_l), \tag{130}$$

where every term in this polynomial in $y_i$ is also of degree at least 4.

For $u^\top (Du)^\top (Du)u$, we have

$$u^\top (Du)^\top (Du)u = u_{i,j} u_j u_{i,k} u_k = u_{i,j} u_{i,k} A_{jlm} y_l y_m A_{kst} y_s y_t, \tag{131}$$

where again, every term in this polynomial in $y_i$ is of degree at least 4.

Let $P(y)$ be a polynomial containing no terms of degree less than 4 (with coefficients being smooth functions on $V_\delta$), and $Q(y) := P(y)/\|y\|^2$. It is easy to see that $\lim_{y \to 0} Q(y) = 0$. And thus $Q(y)$ can be extended continuously to $y = 0$ with $Q(0) = 0$. As $P(y)$ contains no terms whose degree is less than 4, we also have $\lim_{y \to 0} \frac{|Q(y) - Q(0)|}{\|y\|} = 0$. By the definition of differentiability, we have that $Q$ is differentiable at $y = 0$ and $\nabla Q(0) = 0$. Now, for $y \neq 0$, we have $\frac{\partial Q}{\partial y_i} = \frac{P_1(y)}{\|y\|^4}$, where $P_1(y)$ is a polynomial such that every term is of degree at least 5. Thus $\frac{\partial Q}{\partial y_i} \to 0$ as $y \to 0$. This shows that $Q$ is $C^1$. $\qquad\square$

*Proof of Proposition 3.* This denominator of Equation 126 is $\|y\|^2 + \|u\|^2 = \|y\|^2(1 + \frac{u^\top u}{y^\top y})$. Since $\frac{u^\top u}{y^\top y} \to 0$ as $y \to 0$, there is an open set around $y = 0$ where $1 + \frac{u^\top u}{y^\top y} \neq 0$. Thus for any $y$ in this open set such that $y \neq 0$, $\nabla G$ is well-defined.

The squared-norm of the numerator of Equation 126 is

$$\|y + (Du)u\|^2 = (y + (Du)u)^\top (y + (Du)u) = y^\top y \left(1 + \frac{2y^\top (Du)u}{y^\top y} + \frac{u^\top (Du)^\top (Du)u}{y^\top y}\right). \tag{132}$$

Since, by Claim 5, both $\frac{2y^\top (Du)u}{y^\top y}$ and $\frac{u^\top (Du)^\top (Du)u}{y^\top y}$ approach 0 as $y \to 0$, there is an open set around $y = 0$ such that $\|y + (Du)u\|^2 \neq 0$ in this open set when $y \neq 0$. This means that there exists an open set $W$ around $y = 0$ such that $\nabla G$ is well-defined and non-vanishing on $W \setminus \{0\}$. $\qquad\square$

As we are only interested in the behaviour of $A_\delta$ when $\delta \to -\infty$, we can choose a specific $\delta'$ small enough such that $V_{\delta'} \subset W$. Now, for any $\delta < \delta'$, we have $V_\delta \subset V_{\delta'}$, given by the sub-level sets of $G$. Now choose $\delta_{a'}, \delta_a, \delta_b, \delta_{b'}$ such that $\delta_{a'} < \delta_a < \delta < \delta_b < \delta_{b'} < \delta'$. Now, define the vector field $X := \psi(y) \frac{\nabla G}{\|\nabla G\|^2}$, where $\psi$ is a smooth function satisfying

$$\psi(y) = \begin{cases} 1 & \text{if} \quad \delta_a \leq G(y) \leq \delta_b \\ 0 & \text{if} \quad G(y) < \delta_{a'} \text{ or } G(y) > \delta_{b'} \end{cases} \tag{133}$$

and $0 \leq \psi(y) \leq 1$. Then $X$ is a smooth vector field that vanishes outside a compact set, which implies that $X$ is a complete vector field. This means that integrating $X$ gives rise to a one-parameter group of differomophisms $\phi_t : \mathbb{R}^d \to \mathbb{R}^d$ such that $\dot{\phi}_t = X$.

Now consider the map $t \mapsto G(\phi_t(y))$. We have

$$\frac{\partial}{\partial t} G(\phi_t(y)) = \nabla G \cdot \dot{\phi}_t(y) = \nabla G \cdot \left(\psi(\phi_t(y)) \frac{\nabla G}{\|\nabla G\|^2}\right) = \psi(\phi_t(y)). \tag{134}$$

As $\psi(\phi_t(y))$ is always between 0 and 1, the map $t \mapsto G(\phi_t(y))$ is a non-decreasing function of $t$.

**Claim 6.** *If $t \geq 0$, then*

$$G(y) \leq G(\phi_t(y)) \leq G(y) + t. \tag{135}$$

*Proof.* We have

$$G(\phi_t(y)) - G(y) = G(\phi_t(y)) - G(\phi_0(y)) = \int_0^t \frac{d}{d\tau} G(\phi_\tau(y)) d\tau = \int_0^t \psi(\phi_\tau(y)) d\tau. \tag{136}$$

Note that $0 \leq \psi \leq 1$. Thus for $t \geq 0$, we have $0 \leq G(\phi_t(y)) - G(y) \leq t$. $\qquad\square$

**Claim 7.** *Let $y$ be such that $G(y) = \delta \in [\delta_a, \delta_b]$. Then for $t \in [0, \delta_b - \delta]$, we have $G(\phi_t(y)) = G(y) + t$.*

*Proof.* By Claim 6, if $t \in [0, \delta_b - \delta]$, we have $G(\phi_t(y)) \in [G(y), G(y) + t] \subset [\delta_a, \delta_b]$. This means that $\psi(\phi_t(y)) = 1$ for $t \in [0, \delta_b - \delta]$. Thus we can replace the inequality with equality:

$$G(\phi_t(y)) - G(y) = \int_0^t \psi(\phi_\tau(y)) \mathrm{d}\tau = \int_0^t \mathrm{d}\tau = t. \tag{137}$$

$\square$

Now, for any $\delta \in [\delta_a, \delta_b]$, we define $\Phi_\delta := \phi_{\delta - \delta_a}$.

**Proposition 4.** *$\Phi_\delta$ is a diffeomorphism from $V_{\delta_a}$ to $V_\delta$.*

*Proof.* Let $y \in V_{\delta_a}$, i.e. $G(y) < \delta_a$. By virtue of Claim 6, $G(\Phi_\delta(y)) = G(\phi_{\delta - \delta_a}(y)) \leq G(y) + \delta - \delta_a < \delta$. Thus $\Phi_\delta(y) \in V_\delta$. Conversely, let $y \in V_\delta$ and let $y' := \phi_{\delta_a - \delta}(y)$, so that $\Phi_\delta(y') = y$. It suffices to show that $y' \in V_{\delta_a}$, i.e. $G(y') < \delta_a$. Assume the contrary, $G(y') \geq \delta_a$. Note that $G(y') \leq \delta$ because $t \mapsto G(\phi_t(y))$ is non-decreasing. Then we have

$$G(y) = G(\Phi_\delta(y')) = G(\phi_{\delta - \delta_a}(y')) \geq G(\phi_{\delta - G(y')}(y')) = G(y') + \delta - G(y') = \delta, \tag{138}$$

which is a contradiction. Here we used that $t \mapsto G(\phi_t(y))$ is a non-decreasing function for the inequality, and Claim 7 for the second last equality. $\square$

Proposition 4 establishes that $\{\Phi_\delta : \delta \in [\delta_a, \delta_b]\}$ is a family of diffeomorphisms from a fixed domain $V_{\delta_a}$ to $V_\delta$. We can finally invoke the Leibniz integral rule from Equation 124 with $\dot{\Phi} = \frac{\nabla G}{\|\nabla G\|^2}$ on $\partial V_\delta$. Note that

$$\nabla G = \frac{y + (Du)u}{\|y\|^2 + \|u\|^2}, \qquad \|\nabla G\|^2 = \frac{\|y + (Du)u\|^2}{(\|y\|^2 + \|u\|^2)^2}, \qquad \frac{\nabla G}{\|\nabla G\|^2} = H(y)(y + (Du)u), \tag{139}$$

where $H(y) := \frac{\|y\|^2 + \|u\|^2}{\|y + (Du)u\|^2}$. Note that as of now, $H(y)$ is defined on $\partial V_\delta$. We would like to extend $H(y)$ to $V_\delta$. We can rewrite $H(y)$ as

$$H(y) = \frac{y^\top y + u^\top u}{y^\top y + 2y^\top (Du)u + u^\top (Du)^\top (Du)u} = \frac{1 + \frac{u^\top u}{y^\top y}}{1 + \frac{y^\top (Du)u}{y^\top y} + \frac{u^\top (Du)^\top (Du)u}{y^\top y}}. \tag{140}$$

Thanks to Claim 5, $H(y)$ can be extended to $V_\delta$ as a $C^1$ function with $H(0) = 1$. This means that $\frac{\nabla G}{\|\nabla G\|^2}$ can be extended to $V_\delta$ as a $C^1$ vector field.

Now we apply the generalized Stokes' theorem, followed by the definition of divergence, to get

$$A_\delta = \int_{\partial V_\delta} \iota_{\dot{\Phi}} \mathrm{d}y = \int_{\partial V_\delta} \iota_{\frac{\nabla G}{\|\nabla G\|^2}} \mathrm{d}y = \int_{V_\delta} \mathrm{d}\iota_{\frac{\nabla G}{\|\nabla G\|^2}} \mathrm{d}y = \int_{V_\delta} \mathrm{div} \, \frac{\nabla G}{\|\nabla G\|^2} \mathrm{d}y. \tag{141}$$

It suffices to calculate $\mathrm{div} \, \frac{\nabla G}{\|\nabla G\|^2} = \mathrm{div} \, H(y)(y + (Du)u)$. Using Einstein summation notation, we have

$$\mathrm{div} \, H(y)(y + (Du)u) = [H(y)(y_i + u_{j,i}u_j)]_{,i} = H(y)(d + u_{j,ii}u_j + u_{j,i}u_{j,i}) + H_i(y)(y_i + u_{j,i}u_j), \tag{142}$$

which is continuous on $V_\delta$. At $y = 0$, we have

$$\left( \mathrm{div} \, \frac{\nabla G}{\|\nabla G\|^2} \right)(0) = H(0)(d + 0 + 0) + H_i(0)(0 + 0) = d. \tag{143}$$

Thus for $\epsilon > 0$, there exists $\kappa' > 0$ such that if $\|y\| < \kappa'$, then $\left| (\mathrm{div} \, \frac{\nabla G}{\|\nabla G\|^2})(y) - d \right| < \epsilon$. Choose $K''$ such that $V_\delta \subset B_d(0, \kappa')$ for $\delta < K''$. Then

$$|A_\delta - d \cdot Vol(V_\delta)| \leq \int_{V_\delta} \left| \left( \mathrm{div} \, \frac{\nabla G}{\|\nabla G\|^2} \right)(y) - d \right| \mathrm{d}y < \epsilon Vol(V_\delta). \tag{144}$$

This means that

$$(d - \epsilon)Vol(V_\delta) < A_\delta < (d + \epsilon)Vol(V_\delta). \tag{145}$$

Combining Equation 123 and Equation 145, we have

$$\frac{(p(0) - \epsilon)(d - \epsilon)}{p(0) + \epsilon} < \frac{\dfrac{\partial}{\partial \delta} \displaystyle\int_{V_\delta} p(y)\sqrt{g(y)}\mathrm{d}y}{\displaystyle\int_{V_\delta} p(y)\sqrt{g(y)}\mathrm{d}y} < \frac{(p(0) + \epsilon)(d + \epsilon)}{p(0) - \epsilon} \tag{146}$$

for $\delta < K = \min(K', K'')$. Letting $\epsilon \to 0$, we have $\delta \to -\infty$, which entails Equation 116 and thus concludes the proof of Theorem 2. $\qquad\square$

