# OpenReview forum: "On Convolutions, Intrinsic Dimension, and Diffusion Models"
_TMLR — Accepted by TMLR_

### Review · Reviewer_LhHr · 2025-07-09

**Summary Of Contributions:**

This paper provides a rigorous theoretical foundation for the Fast Local Intrinsic Dimension (FLIPD) estimator of local intrinsic dimension (LID), a method introduced in prior work (Kamkari et al., 2024b) for estimating the dimensionality of the submanifold a given data point $x$ belongs to using diffusion models (DMs). Specifically, the authors prove that the key identity underlying FLIPD, which connects LID to the rate of change of the log density under convolution with Gaussian noise, holds under general and realistic manifold assumptions, as opposed to the highly restricted affine case previously assumed in Kamkari et al. (2024b). Additionally, they show an analogous result for uniform convolutions, which provides formal justification for older ball-probability-based estimators as well. The paper is entirely theoretical and contributes to closing the gap between theory and practice for LID estimation via diffusion models.

**Audience:**

Yes

**Broader Impact Concerns:**

The work is largely theoretical and focuses on LID estimation via generative models. I do not think a detailed broader impact discussion is needed.

**Claims And Evidence:**

Yes

**Requested Changes:**

I have highlighted my requested changes below and have specified for each whether they are critical for securing my recommendation for acceptance:

(1) Please include a short discussion or bound on how FLIPD deviates from LID when the learned score function $\hat{s}$ differs from the true score $s$. Even a heuristic discussion or an inequality involving $||\hat{s}-s||$ would help clarify the practical limitations of FLIPD (important but not critical for securing acceptance).

(2) A toy empirical example (e.g., estimating LID on a 2D manifold in $\mathbb{R}^3$) would go a long way toward illustrating the main ideas and convergence properties (not critical for securing acceptance, but would improve the paper).

(3) I have highlighted some minor typos and corrections below:

- Section 2, second paragraph: "is given by (Särkkä & Solin, 2019)" -> please use an in-text citation here, \citet (instead of \citep)

**Strengths And Weaknesses:**

## Strengths

1. The paper tackles a clear and well-motivated gap in the literature: while FLIPD (Kamkari et al., 2024b) performs well empirically, its theoretical foundations were previously limited to affine manifolds. The authors successfully extend this to general submanifolds and disjoint unions thereof, under mild assumptions.

2. The uniform convolution result is a nice result. While not directly useful for DMs, it strengthens the intuition behind older ball-based LID estimators, and could be of interest to the broader theory community.

## Weaknesses

1. The results assume access to the exact score function; in practice, this is always learned approximately via a neural network. While the authors mention this limitation, the paper would be considerably stronger with at least a first-order bound quantifying the bias introduced by imperfect score learning (e.g., some bound on the deviation of FLIPD from LID as a function of $||\hat{s}-s||$).

2. The paper is purely theoretical and includes no empirical results. Even a small illustrative experiment (e.g., convergence behavior of FLIPD as $\delta \rightarrow -\infty$ on a toy manifold) would help ground the theory in practice and potentially shed light on rate of convergence or finite $\delta$ effects.

## Verdict

This paper makes a meaningful theoretical contribution to the literature on LID estimation and generative models, in particular by completing the theoretical justification for the FLIPD estimator under realistic assumptions. While the results are formal in nature and the scope is limited to idealized score access, I believe the work is valuable and the claims made in the paper are well-supported. Moreover, I believe this paper is relevant to at least some portion of TMLR's audience, especially those working on diffusion models, manifold learning, and geometric statistics. As a consequence, I recommend an accept rating for this paper pending the completion of the few minor revisions requested below.

---

> ### Author Response · Authors · 2025-08-11
> **Rebuttal**
>
> Thank you for your review, we appreciate the time you spent on our work. Below we address the points you brought up in your review, please let us know if you would like to further discuss any point. We have also updated our manuscript to include these changes, which are highlighted in blue. These changes include a correction of the typo you found.
>
> > quantifying the bias introduced by imperfect score learning (e.g., some bound on the deviation of FLIPD from LID as a function of $\|\hat{s} - s\|$ .
>
> Note that if we had error bounds between $s$ and $\hat{s}$, and between $\nabla s$ and $\nabla \hat{s}$, then Equation 15 and the triangle inequality would trivially provide an error bound on the LID estimate obtained from FLIPD. While we are aware of work in learning theory providing finite-sample bounds for the discrepancy between $s$ and $\hat{s}$, we are not aware of any work bounding the error between the corresponding Jacobians. If such a bound was to be found, an error bound on LID would immediately follow.
>
> > A toy empirical example
>
> Please note that the main purpose of our work is to theoretically justify an existing LID estimator, FLIPD. The original FLIPD paper already includes these toy examples, and replicating these results is not one of our objectives in this work.

---

### Review · Reviewer_RAVW · 2025-07-18

**Summary Of Contributions:**

This paper provides a theoretical foundation for the FLIPD estimator, which estimates the local intrinsic dimension (LID) of data modeled by diffusion models. While previous work showed FLIPD’s empirical success and proved its correctness only under the restrictive assumption of affine submanifolds, this work extends the analysis to more realistic nonlinear manifold settings. The authors rigorously establish the validity of FLIPD in these general cases and further show that similar results hold when Gaussian convolutions are replaced with uniform ones. These results bridge the gap between theory and practice, enhancing the trustworthiness of LID-based methods in applications like outlier and adversarial detection.

Based on the following comments, the reviewer suggests a revision for this paper.

**Audience:**

Yes

**Broader Impact Concerns:**

No ethical concerns. The paper focuses on the theoretical analysis of existing LID estimators in diffusion models, with no direct implications for sensitive applications. The Broader Impact Statement, if present, adequately addresses the scope and limitations of the work.

**Claims And Evidence:**

Yes

**Requested Changes:**

There are some concerns and questions about the technical details in this paper.

**1.**  Eq. (2) about the probability density function (pdf) of an isotropic Gaussian is confusing. Here, $x \in \mathbb{R}^D$ and mean $\mu \in \mathbb{R}^d$ are vectors, while the covariance $\delta$ in $\mathcal{N}_D(x; \mu, \delta)$ is a scalar. Moreover, in Eq. (2), where do the terms $\exp(-2\delta)$ and $\exp(-D\delta)$ come from?

**2.** The major contribution of this paper is a rigorous theoretical justification that Equation (1) remains valid when the data distribution $p$ is supported on a general disjoint union of smooth submanifolds. This result significantly generalizes previous work, which was limited to affine submanifolds, and formally supports the use of FLIPD for estimating local intrinsic dimension (LID) in more realistic geometric settings (see Theorems 1 and 2). However, a key limitation is that the paper does not provide guidance on how the derived theoretical results can be practically applied to compute LID in real-world diffusion models. In particular, it would be valuable to include discussion or examples showing how these results translate into usable estimators in practice, especially in the context of high-dimensional generative models.

**3.** Building on the concern raised above, it would greatly strengthen the paper if the authors included empirical experiments demonstrating how the derived equation can be leveraged to compute the local intrinsic dimension (LID) in practice.

It is believed that the paper would be accepted if the authors could fix the above points.

**Strengths And Weaknesses:**

**1.**  The paper makes a significant theoretical contribution by rigorously proving the correctness of the FLIPD estimator for local intrinsic dimension (LID) when the data distribution is supported on a disjoint union of general smooth submanifolds. This removes the strong affine submanifold assumption from prior work and provides a much broader foundation for the estimator’s applicability.

**2.** The paper extends the analysis from Gaussian to uniform convolutions, suggesting that the theoretical framework may be robust to the choice of smoothing kernel. This generalization could open up new directions for understanding the geometry of data distributions under different generative processes.

**3.** The paper is well written and clearly organized, making technically involved results accessible to readers familiar with the field. The authors effectively communicate the motivation, main results, and implications of their work.

Please refer to the following section for main weaknesses.

---

> ### Author Response · Authors · 2025-08-11
> **Rebuttal**
>
> Thank you for your review, we appreciate the time you spent on our work. Below we address the points you brought up in your review, please let us know if you would like to further discuss any point. We have also updated our manuscript to include these changes, which are highlighted in blue.
>
>
> > Eq. (2) about the probability density function (pdf) of an isotropic Gaussian is confusing.
>
> Please note that $\delta \in \mathbb{R}$ is not a covariance matrix, it is just a scalar. Since the Gaussians we consider all have covariance matrix given by $e^{2\delta}I_D$, we denote their densities as $\mathcal{N}(x; \mu, \delta)$. Please note as well that $x$ and $\mu$ are both $D$-dimensional vectors. All the terms in equation 2 correspond to the pdf of a multivariate Gaussian with mean $0$ and covariance $\sigma^2 I_D$ in $\mathbb{R}^D$, that is $\dfrac{1}{(2\pi)^{D/2}} \dfrac{1}{\sigma^D} \exp(\dfrac{1}{2\sigma^2} \Vert x - \mu \Vert^2)$; when parameterizing the density using the log standard deviation $\delta$ rather than the standard deviation $\sigma$, we have $\sigma = e^{\delta}$, yielding the terms you asked about.
>
> > In particular, it would be valuable to include discussion or examples showing how these results translate into usable estimators in practice, especially in the context of high-dimensional generative models
>
> In practice, FLIPD estimates LID by plugging-in the learned score function $\hat{s}$ into equation 16. Note that our work is not meant to provide a change to how FLIPD operates in practice. FLIPD is existing work, and our goal is simply to formally justify this estimator, i.e. to explain why it works in practice.
>
> > empirical experiments demonstrating how the derived equation can be leveraged to compute the local intrinsic dimension (LID) in practice
>
> Please note that the original FLIPD paper included empirical experiments. We emphasize once again that our goal is to provide a formal justification of why those experiments showed good performance, not to reproduce these experiments.

---

> ### Comment · Reviewer_RAVW · 2025-09-15
>
> Thank you for the clarification. The responses address my questions. I recommend acceptance for this paper.

---

### Review · Reviewer_5wyd · 2025-08-10

**Summary Of Contributions:**

This paper aims to provide a theoretical justification for local intrinsic dimension (LID) estimation, clarifying the validity of recent diffusion-based dimension estimation techniques, such as FLIPD.
The main problem addressed is whether the derivative of the log of a Gaussian-noise–convolved probability density remains correlated with the local intrinsic dimension when the support is a general embedded submanifold (rather than an affine subspace, as in Eq. 1), and whether an analogous result holds when the data is convolved with uniform noise (Eq. 9).

The key technical approach for extending the analysis from affine subspaces to embedded submanifolds is a two-step strategy:
1. Use the definition of a $d$-dimensional embedded submanifold to pull back the calculations to a $d$-dimensional Euclidean space.
2. Apply appropriate bounding techniques in this reduced space.

For the Gaussian noise case, the authors compute the limiting probability (Prop. 1) and the expected squared norm (Prop. 2) for manifold-supported data. For the uniform noise case, the second step is adapted to bound the “area” of the annulus formed by intersecting the $d$-dimensional submanifold with two balls, reducing to a ring in the affine case. The proofs naturally extend to union of disjoint submanifolds, since the analysis considers a localized limit.

Overall, the paper establishes mathematically rigorous results for problems that are intuitive but technically challenging in manifold dimension estimation.

**Audience:**

Yes

**Claims And Evidence:**

Yes

**Requested Changes:**

### Requested Changes

1. It may be helpful to discuss related recent work that also examines low-dimensional or low-rank structures in diffusion models, for example:
   a. Wang et al., *Diffusion models learn low-dimensional distributions via subspace clustering* ([arXiv:2409.02426](https://arxiv.org/abs/2409.02426), 2024)
   b. Chen et al., *Exploring low-dimensional subspace in diffusion models for controllable image editing* (NeurIPS 2024)

2. While the main contributions are theoretical, adding a small numerical experiment or visualization could make the results more accessible to a broader audience and help illustrate the intuition behind the proofs (see Weakness 1).

### Questions for the Authors
1. The proofs proceed by isolating the $d$-dimensional term and showing the residual log derivative vanishes. Is this behavior specific to Gaussian and uniform noise, or might Theorems 1–2 extend to a broader class of noise distributions, e.g., sub-Gaussian?
2. How should one interpret an estimated LID (or FLIPD value) that is non-integer? Does it indicate that the data only approximately lies on a manifold, or that the score function is imperfectly learned?
3. Since the analysis takes the limit $\delta \to -\infty$, equivalent to $t = \lambda^{-1}(e^\delta) \to 0$, does this mean that only the score function $s(\cdot,t)$ at $t=0$ is required for LID estimation?

**Strengths And Weaknesses:**

### Strengths
1. General data assumptions and mild regularity conditions.
2. Rigorous, well-structured proofs with clear logical flow.
3. Consideration of non-Gaussian (uniform) noise, which is relevant in prior works such as Bansal et al. *Cold diffusion: Inverting arbitrary image transforms without noise.* (NeurIPS 2023)

### Weaknesses
1. No numerical validation or illustrative figures.
2. Limited discussion of practical implications. For instance, practitioners might expect some characterization of the differences between Gaussian and uniform noise (given their distinct spectral behaviors), but the paper only establishes that both are valid for dimension estimation.

---

> ### Author Response · Authors · 2025-08-11
> **Rebuttal**
>
> Thank you for your review, we appreciate the time you spent on our work. Below we address the points you brought up in your review, please let us know if you would like to further discuss any point. We have also updated our manuscript to include these changes, which are highlighted in blue. These changes include a discussion of the related works you brought up.
>
> > No numerical validation or illustrative figures.
>
> While we agree that a figure conveying the intuition of why our results are true would be helpful, we found it difficult to produce a figure conveying this intuition. As for the lack of numerical experiments, we highlight that the main goal of our work is simply to formally justify FLIPD, an existing estimator of LID. The original FLIPD paper includes extensive numerical validation, and it is not our objective to replicate the results in the original FLIPD paper.
>
> > Limited discussion of practical implications. For instance, practitioners might expect some characterization of the differences between Gaussian and uniform noise (given their distinct spectral behaviors), but the paper only establishes that both are valid for dimension estimation.
>
> Could you please clarify what you mean by this?
>
> > The proofs proceed by isolating the d-dimensional term and showing the residual log derivative vanishes. Is this behavior specific to Gaussian and uniform noise, or might Theorems 1–2 extend to a broader class of noise distributions, e.g., sub-Gaussian?
>
> As we briefly mention in the conclusions section, we do believe that our result could be further extended. However, we do not know exactly what conditions might be required to formally extend the result. We hypothesize that a decomposition like those of equations 23 or 43 would be needed, along with some tail condition.
>
> > How should one interpret an estimated LID (or FLIPD value) that is non-integer? Does it indicate that the data only approximately lies on a manifold, or that the score function is imperfectly learned?
>
> This will typically be the case: first, FLIPD (equation 16) requires setting some very negative $\delta_0$ rather than taking the formal limit as $\delta \rightarrow -\infty$, which can result in non-integer estimates. Second, approximation error between the true and learned score function can result in non-integer limits if the learned score function does not correspond to a proper score function (i.e. if it is not the gradient of some log density). If scalar estimates are highly important, one could always round FLIPD to its nearest integer, but we also highlight that FLIPD not always producing integer estimates has advantages as well. For example, the original FLIPD paper highlights advantages from FLIPD being differentiable.
>
> > Since the analysis takes the limit $\delta \rightarrow -\infty$, equivalent to $t=\lambda^{-1}(e^\delta)\rightarrow 0$, does this mean that only the score function $s(\cdot, t)$ at $t=0$ is required for LID estimation?
>
> This is almost correct: note that FLIPD requires computing $s$, but also $\nabla s$ (equation 15), and it cares about the limit of these quantities as $t \rightarrow 0$. Thus, we care about the score function in a vicinity of 0. More formally, for any $\epsilon >0$, only $s(\cdot, t)$ for every $t \in (0, \epsilon)$ is needed for LID estimation.

---

> > ### Comment · Reviewer_5wyd · 2025-09-23
> >
> > Thank you for the clarifications. Although verification for the Uniform noise case is still missing, the paper is technically solid, and I have recommended acceptance.

---

> > > ### Author Response · Authors · 2025-09-30
> > > **On experiments for the uniform case**
> > >
> > > Thank you very much for recommending acceptance, we are glad you found our clarifications satisfactory.
> > >
> > >
> > > As for experiments for the uniform case, please note that our work is not meant to enable a FLIPD-like LID estimator when uniform noise is used. The reason for this is that, in the Gaussian case, diffusion models essentially train a denoiser which removes the Gaussian noise added during the forward process. This denoiser is useful for FLIPD only thanks to the time reversal result of SDEs (eq 11), which links the learned denoiser to the score function (which is in turn related to the convolution of interest). To the best of our knowledge, there is no analogous result for uniform noise. In turn, even if we were to train a denoiser which removes added uniform noise from noisy data, there would be no explicit way of linking our trained denoiser to the convolution of interest, and thus no FLIPD-like LID estimator would follow. In short, our result with uniform noise is of interest because it involves quantities that naturally appear in ML and because it formally justifies older LID estimators (which are not based on diffusion-like models), but it does not immediately provide an alternative to FLIPD when uniform, instead of Gaussian, noise is used. We will further clarify this point in the camera-ready version of our work.

---

### Decision · Action_Editor_kECk · 2025-09-27

**Recommendation:** Accept as is

**Audience:**

Yes

**Audience Explanation:**

The study is valuable to people in understanding diffusion models and intrinsic dimension of image dataset.

**Claims And Evidence:**

Yes

**Claims Explanation:**

The paper’s core claims are backed by rigorous proofs extending FLIPD from affine to nonlinear manifolds and by parallel results under both Gaussian and uniform convolutions, which strengthens robustness and relevance. This directly addresses the prior gap between empirical success and theory, lending the method credible foundations for applications like outlier/adversarial detection. For full confidence, clarity on assumptions (manifold regularity, sampling/noise conditions) and brief empirical sanity checks would seal the case.

All the reviewers agree on this point, and this is a valuable contribution both theoretically and empirically.